# Proteomic atlas of organ vasculopathies triggered by Staphylococcus aureus sepsis

Alejandro Gómez Toledo[1,2,12], Gregory Golden[1,2,12], Alexandre Rosa Campos[3], Hector Cuello [4], James Sorrentino [5], Nathan Lewis [5,6,7], Nissi Varki[2,8], Victor Nizet [9,10], Jeffrey W. Smith[11] & Jeffrey D. Esko [1,2]*

Sepsis is a life-threatening condition triggered by a dysregulated host response to microbial infection resulting in vascular dysfunction, organ failure and death. Here we provide a semi-quantitative atlas of the murine vascular cell-surface proteome at the organ level, and how it changes during sepsis. Using in vivo chemical labeling and high-resolution mass spectrometry, we demonstrate the presence of a vascular proteome that is perfusable and shared across multiple organs. This proteome is enriched in membrane-anchored proteins, including multiple regulators of endothelial barrier functions and innate immunity. Further, we automated our workflows and applied them to a murine model of methicillin-resistant *Staphylococcus aureus (MRSA)* sepsis to unravel changes during systemic inflammatory responses. We provide an organ-specific atlas of both systemic and local changes of the vascular proteome triggered by sepsis. Collectively, the data indicates that MRSA-sepsis triggers extensive proteome remodeling of the vascular cell surfaces, in a tissue-specific manner.

[1] Department of Cellular and Molecular Medicine, University of California, San Diego, La Jolla, CA, USA. [2] Glycobiology Research and Training Center, University of California, San Diego, La Jolla, CA, USA. [3] Proteomics Core Facility, Sanford-Burnham-Prebys Medical Discovery Institute, La Jolla, CA, USA. [4] Molecular Oncology Laboratory, Quilmes National University, Buenos Aires, Argentina. [5] Bioinformatics and Systems Biology Graduate Program, University of California, San Diego, La Jolla, CA, USA. [6] Departments of Pediatrics and Bioengineering, University of California, San Diego, La Jolla, CA, USA. [7] Novo Nordisk Foundation Center for Biosustainability, University of California, San Diego, La Jolla, CA, USA. [8] Department of Pathology, University of California, San Diego, La Jolla, California, USA. [9] Collaborative to Halt Antibiotic-Resistant Microbes (CHARM), Department of Pediatrics, University of California, San Diego, La Jolla, CA, USA. [10] Skaggs School of Pharmacy and Pharmaceutical Sciences, University of California, San Diego, La Jolla, CA, USA. [11] The Cancer Center and The Inflammatory and Infectious Disease Center, Sanford-Burnham-Prebys Medical Discovery Institute, La Jolla, CA, USA. [12] These authors contributed equally: Alejandro Gómez Toledo, Gregory Golden. *email: jesko@ucsd.edu

The mammalian circulatory system is built of specialized tissues capable of specifying distinct vascular environments. Due to anatomical and histological constraints, the structure, morphology, and composition of the vasculature vary across different organs[1,2]. For example, the highly specialized blood–brain barrier (BBB) is built of continuous and rather impermeable brain capillaries, whereas the liver sinusoids or the kidney glomeruli are made of more permeable, discontinuous, or fenestrated blood vessels. The endothelial glycocalyx is a cell surface layer located at the luminal side of the blood vessels, and is mainly composed of glycans, glycolipids, glycoproteins, and proteoglycans[3]. The glycocalyx modulates organ-specific functions, vascular hemostasis, and multiple aspects of innate immunity[4].

Rapid remodeling of the vascular surfaces occurs during systemic inflammatory responses and sepsis[5,6], with increased degradation and shedding of glycocalyx components. Shed material can fuel dysregulated inflammatory loops, acting as damage-associated molecular patterns (DAMPs) or delaying glycocalyx reconstitution by blunting growth factor signaling[7]. Remodeling of the endothelial cell surface occurs during leukocyte recruitment and extravasation, and leads to activation of the coagulation and complement systems[8–11]. Infiltrating immune cells and soluble plasma proteins also modulate the final make-up of the endothelial glycocalyx.

The occurrence of multiple organ failure (MOF) is a cardinal sign of severe sepsis or "septic shock"[12–14]. However, the molecular mechanisms determining why certain organs are more prone to failure than others are not fully understood. Given its location between the blood and tissue compartments, the vascular glycocalyx is an attractive target for therapeutic intervention to prevent MOF in sepsis. In fact, it is possible that different vascular beds may respond differently to septic factors since they are equipped with unique and dynamic cell surface proteomes and glycomes. Additionally, molecular remodeling of the vascular surfaces could also facilitate acute and adaptive immune responses. The ability to track these processes in vivo with temporal and spatial resolution is key to understanding early events during systemic inflammatory responses. Eventually, such insights might facilitate the molecular classification of sepsis subtypes based on specific pathogens and/or host vascular responses.

Unfortunately, there is limited knowledge of the molecular composition of the vascular glycocalyx in vivo, its variation across different organs, and the changes that occur during disease. Most studies reported to date rely either on specialized fixation and staining techniques, or on indirect assays, for example tracking the levels of selected markers of endothelial dysfunction in plasma[15–17]. Proteomic approaches have also demonstrated changes in the glycocalyx of cultured endothelial cells after exposure to proinflammatory agents[18,19]. However, it is most likely that cell-based systems do not fully recapitulate the complex nature of the in vivo vascular glycocalyx due to the absence of perivascular cells and plasma components. In fact, ~40% of the proteins expressed on luminal endothelial cell plasma membranes isolated from rat lungs are totally absent in cultured rat lung microvascular endothelial cells, documenting the inadequacy of cell culture models to duplicate the natural environment of endothelial cells[20].

Terminal systemic perfusion of animals has been used to deliver reactive ester-derivatives of biotin to label cell surface proteins accessible from the vascular flow[21,22]. This approach is simple and effective, and depending on the chemistry of the linkers and the perfusion conditions, it can result in the specific labeling of tumors and identification of tumor biomarkers[23]. The use of biotin facilitates downstream affinity chromatography and can be easily interfaced with shotgun proteomics to dissect the vascular cell surface proteome. Studies have demonstrated the utility of these approaches to profile and quantify vascular antigens in metastatic lesions in the liver and B-cell lymphomas, using time-of-flight (TOF) mass spectrometry[24,25]. However, new generation mass spectrometers with orbitrap-based technology are now widely available, providing increased sensitivity, scan speed, and mass accuracy compared to their predecessors. These instruments facilitate high-resolution measurements of fragment ions, improved proteome coverage, lower false discovery rates, and more robust absolute and semi-quantitative proteome analysis.

In this report, we apply a strategy based on systemic biotinylation of murine tissues, streptavidin affinity chromatography and high-resolution LC–MS/MS, all in a fully automated workflow, achieving high-throughput and reproducibility. The method was employed to generate a label-free semi-quantitative molecular atlas of the murine vascular cell surface proteome at the organ level (liver, kidney, brain, heart, and white adipose tissue) and how it changes during methicillin-resistant *Staphylococcus aureus* (MRSA) sepsis.

## Results

**Systemic labeling of vascular structures in murine organs**. Due to its systemic nature, a septic response is difficult to recapitulate in vitro. Thus, a more universal approach is needed to track proteome changes triggered by a septic insult in vivo. We explored the labeling of murine vascular compartments using terminal systemic perfusion with ester derivatives of biotin to tag, purify, and identify proteins normally exposed to the vascular flow. The labeling conditions are summarized in Fig. 1 and are similar to methods previously reported by Rybak et al.[21]. We subjected wildtype C57BL/6J mice to this procedure using sulfo-NHS-biotin as described in the Methods section, and verified the extent of labeling and localization of biotinylated material.

First, we harvested biotinylated organs, as well as control tissue derived from PBS-perfused animals. Tissues were homogenized, the homogenates run on SDS-PAGE, and tagged proteins were detected by blotting with streptavidin. As shown in Supplementary Fig. 1, multiple protein bands were detected in the biotinylated samples, whereas only faint bands were observed in tissues from animals that were perfused with PBS. Tissue-specific differences were also observed, as suggested by differential mobility and intensity of the biotinylated protein bands detected in kidney and heart. These differences suggested that the accessibility and/or the composition of the vascular proteomes might differ among the organs.

To better resolve the tissue compartments targeted by sulfo-NHS-biotin perfusion, multiple organs were harvested, cryosectioned, and stained with fluorophore-conjugated streptavidin. Histological examination by confocal microscopy showed biotinylated proteins in close association with blood vessels (Fig. 2a–d). For example, strong streptavidin reactivity was detected in the liver around the hepatic central veins and the sinusoidal microvasculature, but not in association with parenchymal hepatocytes (Fig. 2a). In the kidney, biotinylated material was restricted to the glomerular compartments and proximal tubule (Fig. 2b). In the heart (Fig. 2c) and brain (Fig. 2d), streptavidin reactivity localized primarily within the microvasculature, with no obvious penetration into deeper parenchymal regions.

To determine if the biotin labeling was specifically associated with endothelial cell surfaces, co-staining with Isolectin B4 (IB4) was performed (Fig. 2a–d). IB4 is a glycoprotein derived from the African legume *Griffonia simplicifolia* that recognizes terminal

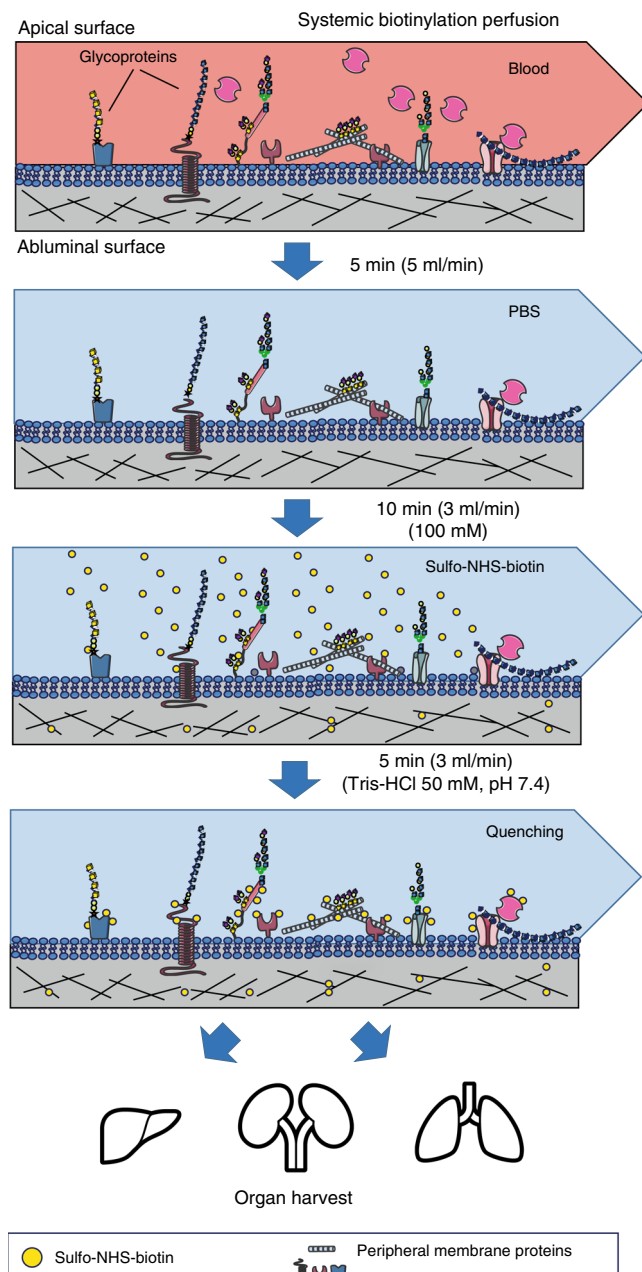

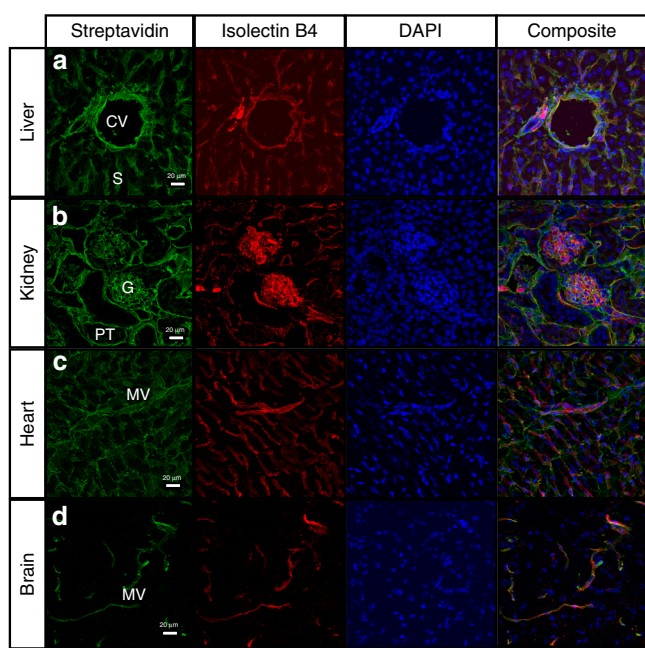

**Fig. 2** Protein biotinylation is primarily associated with vascular compartments. Murine tissues from animals perfused with sulfo-NHS-biotin were excised and subjected to cryosectioning, followed by histological analysis using fluorescently labeled streptavidin. Cryosections from liver (**a**), kidney (**b**), heart (**c**), and brain (**d**) were imaged using confocal microscopy. Most of the streptavidin reactivity was closely associated with vascular tissue structures such as the liver sinusoid or the kidney glomerular microvasculature. Tissue slides were also co-stained with IB4 to visualize the endothelial lumen. Partial co-localization between streptavidin and IB4 stains indicated incorporation of biotin into the endothelial glycocalyx but also in the nearby extracellular matrix and the vascular extracellular space. Histological analysis was conducted in biological triplicates, but only representative slides are shown. CV: hepatic central veins, S: sinusoids, G: glomeruli, PT: proximal tubules, MV: microvasculature. Scale bar, 20 µm

**Fig. 1** Workflow for in vivo biotinylation of vascular antigens. Animals were first perfused with saline (PBS) to remove blood, followed by biotinylation using an isotonic solution of sulfo-NHS-biotin. Unreacted NHS-groups were quenched by perfusion with a Tris-HCl buffer (pH 7.4). All buffers were kept ice-cold and the perfusion times were kept as short as possible to minimize potential tissue damage and disruption. After biotinylation, multiple organs were harvested and preserved for histological analysis, or immediately homogenized and subjected to proteomics analysis

alpha-linked galactose residues on the carbohydrates lining the endothelial lumen. Partial overlap between the streptavidin and the IB4 staining patterns was observed in all tissues, confirming the presence of biotinylated material on the endothelial lumen. However, strong streptavidin reactivity was also detected at the basement membrane, and within the adjacent extracellular matrix (ECM) of the endothelial cells. Intracellular staining remained low, confirming the inability of sulfo-NHS-biotin to penetrate the cell membrane due to the charged character of its sulfate functionality.

**Shared and organ-specific vascular proteomics signatures**. To qualitatively explore the scope of the systemically labeled vascular proteome, two major organs (liver, kidney) were homogenized and subjected to streptavidin enrichment and trypsinization. Peptide digests were analyzed through an online 2D-LC-MS/MS workflow at high/low pH, as described in the Methods section, to perform deep fractionation of the samples and to increase proteome coverage. Briefly, samples were first separated by reverse-phase chromatography on a C18-column at pH 10 and five consecutive fractions were collected by eluting at increasing acetonitrile concentrations (17%, 19.5%, 22%, 26% and 65%). The individual fractions were then separated at pH 3 on a C18-column and analyzed by LC-MS/MS (Fig. 3a, b). Matching control tissue from PBS-perfused animals was subjected to the same analytical procedure to account for background signals (i.e. non-biotinylated proteins that non-specifically interact with the streptavidin beads). Only peptides displaying at least a 2-fold enrichment compared to the PBS background of the respective organ were considered for further analysis. To add more stringency, a positive protein identification required a minimum of two unique peptides with two MS/MS scans each. The complete

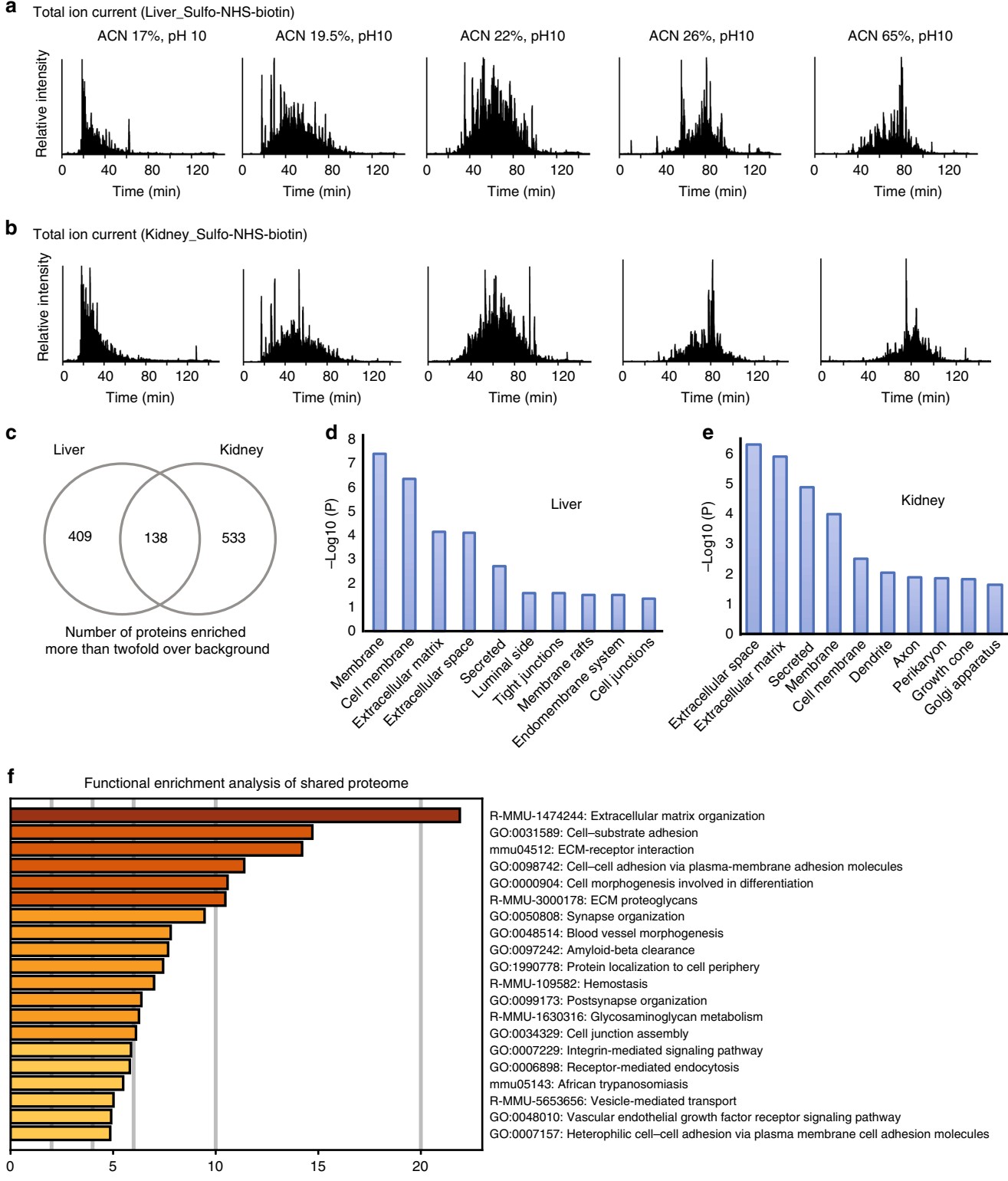

**Fig. 3** Online 2D-LC-MS/MS proteomics analysis of liver and kidney proteins. Biotinylated proteins from one liver and one kidney samples were enriched on streptavidin columns, trypsinized and subjected to an online 2D-LC-MS/MS workflow on a C18 column at high/low pH. Total ion chromatograms for 5 consecutive fractions from liver (**a**) and kidney (**b**) peptide digests are shown, indicating that proteins eluting from the C18 column at high pH and at increasing acetonitrile concentrations display different chromatographic behaviors when separated at low pH, consistent with an orthogonal fractionation strategy. Venn diagrams showing unique and shared protein components (**c**). Hypergeometric enrichment test for subcellular localization showed significant enrichment for proteins located in the cell membrane and extracellular matrix in both liver (**d**) and kidney (**e**). Functional enrichment analysis of shared proteome signatures is shown in (**f**)

bioinformatic treatment of the data was performed as detailed in the Methods section.

After filtering the data, 409 non-redundant proteins were identified in the liver and 533 proteins were detected in kidney samples, at a 1% false discovery rate (Fig. 3c and Supplementary Data 1). A number of proteins (138) were shared between the tissues. We analyzed the subcellular localization of all protein identifications using hypergeometric enrichment tests on their associated Gene Ontology (GO) terms. In both liver and kidney datasets, we found a significant enrichment for proteins associated with the plasma membrane and the extracellular matrix (Fig. 3d, e). This shared proteome included cadherins and cadherin-like proteins, integrins, collagens, laminins, proteoglycans, cellular receptors, and enzymes as well as a large repertoire of receptor-type protein tyrosine phosphatases (Supplementary Data 1). Classical endothelial markers such as VE-cadherin, endoglin, vascular cell adhesion molecule 1 (VCAM1), and intercellular adhesion molecule 1 (ICAM1) were also identified. Functional enrichment analysis of this signature retrieved pathways associated with cell adhesion, vascular morphogenesis, proteoglycan metabolism, and vascular endothelial growth factor (VEGF)-signaling pathways (Fig. 3f).

Unique organ-specific proteomic signatures were also detected. For example, liver samples were characterized by the presence of multiple scavenger receptors such as C-type lectin family 4 F (Clec4F), Scavenger Receptor Class B Member 1 (Scarb1), stabilin 1 (Stab1), Stab2, and asialoglycoprotein receptor 2 (Asgr2), reflecting the filtering functions of the hepatic reticuloendothelial system (Supplementary Data 1). We also found a number of proteins involved in lipoprotein remodeling and clearance, including low density lipoprotein receptor (Ldlr), angiopoietin-related protein 3 (Angptl3), hepatic triglyceride lipase precursor (Lipc), fatty acid binding protein 1 (Fabp1), and LDLR associated protein 1 (Ldlrap1), consistent with the role of the liver as the major target for lipoprotein metabolism and uptake. Clec4F is a marker for Kupffer cells, whereas Angptl3, Lipc, and Ldlrap1 are soluble proteins presumably bound to the glycocalyx. Similarly, kidney samples were specifically enriched in proteins involved in the regulation of blood pressure and fluid balance, including components of the renin-angiotensin system, such as angiotensin converting enzyme (Ace), Ace2, glutamyl aminopeptidase (Enpep), and renin 2 (Ren2).

Finally, a network approach was applied to focus on potential protein-protein associations amongst the unique vascular proteomes. All proteins identified in each tissue were searched through the Search Tool for the Retrieval of Interacting Genes/Proteins (STRING) database to generate a network based on physical and functional interactions. Only high confidence protein–protein associations (STRING association scores >0.07) were retained in the network. The Louvain method was used to identify communities (or clusters) displaying a higher density of interconnected nodes than expected by random chance[26]. These clusters were further segregated via force-directed visualization algorithms, and subjected to functional enrichment analysis. Roughly, 33% of the liver and 31% of the kidney proteins clustered into 4 and 5 distinct Louvain communities, respectively (Supplementary Fig. 2a–i). Each cluster covered several functional layers that were often enriched in distinct biological processes. The results from the Louvain clustering showed a clear pattern of functional commonalities between the biotinylated tissues. Interestingly, some of the clusters were enriched in organ-specific processes such as signaling through receptor tyrosine kinases and glutathione metabolism in the kidney (Supplementary Fig. 2b, e), or regulation of lipid metabolic pathways and lipoprotein particle clearance in the liver (Supplementary Fig. 2h).

## MRSA-sepsis results in profound liver vasculopathy.

Since sepsis is a systemic syndrome of vascular dysfunction, we reasoned that tracking specific molecular changes at the blood/tissue interface could generate new insights into sepsis-associated vasculopathies. To explore this idea, a proteomics workflow was applied to a murine model of sepsis triggered by MRSA bacteremia. Briefly, wildtype C57BL/6J mice were intravenously infected through the retroorbital route with $5 \times 10^7$ colony-forming units (cfu) of MRSA, a model that induces lethality within 48 h post-infection[27], or with PBS as a control. Since changes in the vascular compartments are expected to precede organ damage and lethality, we focused on the early pre-mortality stages of the disease 24 h post-infection[28].

Analysis of the pathogen burden at this time point revealed dramatically different bacterial invasion and/or colonization across tissues. Liver, kidney, and heart samples exhibited the highest pathogen infiltration, followed by brain and WAT (Fig. 4a and Source Data File). Gross pathology of the liver showed white necrotic plaques that were completely absent in control animals (Fig. 4b), in line with previously published findings[29]. To confirm that the hepatic pathogen burden was linked to ongoing liver failure, we measured plasma levels of liver markers alanine aminotransferase (ALT) and aspartate aminotransferase (AST). Indeed, both markers were increased at 24 h post-infection (Fig.4c and Source Data File).

To obtain a global picture of the organ pathologies, multiple infected and uninfected tissues were subjected to histopathological analysis. Examination of hematoxylin and eosin (H&E) stained tissues revealed different degrees of inflammation, bacterial colonization and tissue damage across organs (Supplementary Fig. 3a–j). Liver tissues stood out from the others by showing more leukocyte infiltrates, more bacterial microabscesses and severe signs of necrotic thrombosis (Fig. 4d). Pathological hypercoagulation was characterized by the presence of large thrombi in the major veins of the liver, leading to blood vessel occlusion and disseminated tissue necrosis. Necrotic loci spread out over large areas adjacent to the thrombotic zones, and the thrombi were surrounded by layers of polymorphonuclear cells.

To assess the suitability of the biotinylation perfusions for the analysis of vascular alterations during sepsis, liver cryosections from animals challenged with MRSA and systemic biotinylation, were probed with fluorescently labeled streptavidin. As shown in Fig. 4e, streptavidin reactivity was detected in areas of thrombosis, around the blood vessels and at the surface of infiltrating immune cells. Notably, the thrombi were also intensely stained, whereas regions of tissue necrosis were streptavidin negative, consistent with occluded blood perfusion into those areas.

## Proteomics profiling of vascular surfaces during sepsis.

Given that the biotinylation perfusion was able to label protein targets within these key vasculopathic regions, the proteomics workflow was expected to render a molecular snapshot of the septic organ vasculatures, including the injured hepatic surfaces. For each experiment, 3 MRSA-infected and 3 uninfected mice were anesthetized and subjected to the systemic biotinylation protocol at 24 h post-infection. In addition, one infected and one uninfected mouse were perfused only with saline to account for potential background signals. Five different organs were selected for proteomics analysis based on their clinical relevance: liver, kidney, heart, brain, and white adipose tissue (WAT). In total, we performed four independent experiments resulting in a total of 160 LC-MS/MS runs. The samples were quantified using the MS label-free strategy described above. To minimize variability introduced by manual sample handling, the workflow was fully

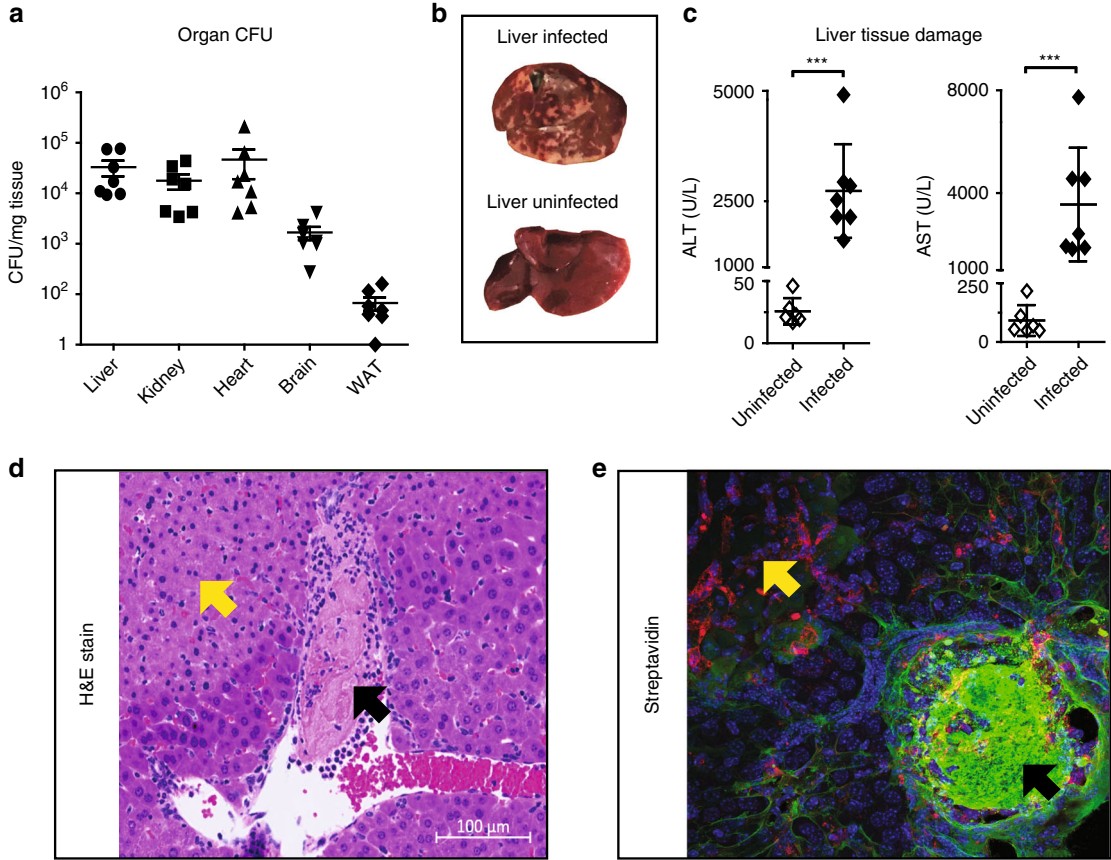

**Fig. 4** The murine model of MRSA-sepsis has a profound liver vasculopathic phenotype. Analysis of the pathogen burden expressed as colony-forming units (CFUs) at 24 h post-infection (**a**). A representative image of liver gross pathology in uninfected and infected mice at 24 h post-infection shows the presence of white areas corresponding to necrotic areas across the liver parenchyma (**b**). Quantification of soluble markers of liver damage alanine aminotransferase (ALT) and aspartate aminotransferase (AST) in serum collected 24 h post-infection (**c**). Hematoxylin and eosin staining of representative liver sections from infected animals showing signs of necrosis (yellow arrow) and thrombosis (black arrows), scale bar, 100 μm. **d** Representative liver section from mice challenged with MRSA, subjected to systemic biotinylation, and stained with streptavidin (green), IB4 (red) and DAPI (blue) (**e**). For **a** and **c**, data was pooled from two independent experiments with $n = 3$ and $n = 4$ mice per experiment, where mice were infected but not subjected to biotinylation perfusion. Data are represented as mean ± SD. ***$p < 0.001$ by two-sided Student's $t$-test

automated by transferring all sample preparations to a BRAVO liquid handling platform. To speed up the process and to be able to perform relative quantification, all samples were analyzed using a single run LC-MS/MS, instead of the 2D-LC-MS/MS approach used for deep fractionation. However, since sepsis is notoriously heterogenous, we also applied strict bioinformatic and statistical criteria to only focus on robust and reproducible changes, as specified in the method section.

The proteomics screening identified robust and reproducible alterations of a few hundred of proteins from each individual organ (liver: 272, kidney: 300, heart: 275, WAT: 185, and brain: 85), according to a two-way analysis of variance (ANOVA) with a permutation-based false discovery rate correction for multiple test comparisons. The relative protein quantification for each organ is summarized in Supplementary Data 2. Missing values were addressed by requiring a cut-off corresponding to 75% valid values in at least one group (infected + biotin, uninfected + biotin, or PBS controls). Profile plots of proteins complying to strict criteria (see Methods) are shown in Supplementary Fig. 4a–e. In general, the identified proteome changes covered a wide dynamic range. Notable interassay and intraassay variability was also observed across the samples as judge by profile plots of their Label-Free Quantification (LFQ)- normalized intensities (Supplementary Fig. 4a–e, upper panels). Nevertheless, plotting the LFQ-profiles of the top 10 proteins ranked by fold change

(Supplementary Fig. 4a–e, lower panels), showed a clear separation between infected and non-infected samples in all tissues, indicating that the method can detect quantitative differences triggered by sepsis.

Closer inspection of the LFQ-normalized intensities showed strong correlations for replicates in the same group. For example, proteins identified across three infected livers, displayed high Pearson correlation coefficients ($r = 0.93$, $0.96$ and $0.96$) (Supplementary Fig. 5a), whereas correlations decreased when comparing infected vs non-infected controls ($r = 0.91$, $0.84$, and $0.77$) (Supplementary Fig. 5b). Even lower correlations were observed when comparing biotinylated samples with non-labeled PBS controls, independently of their infection status (infected vs. PBS, $r = 0.62$, $0.69$, and $0.62$; uninfected vs. PBS, $r = 0.56$, $0.71$, and $0.74$) (Supplementary Fig. 5c, d). Similar results were found in other tissues indicating that the identity of the proteome accessible through systemic biotinylation differs dramatically from uninfected and PBS-perfused tissues. More importantly, these findings also indicate that the methodology captures differences specifically associated with infection.

**Global and organ-specific proteome changes**. A comparative view of the proteomic changes taking place across the five organs revealed a consistent organ-specific hierarchy (Fig. 5). In liver,

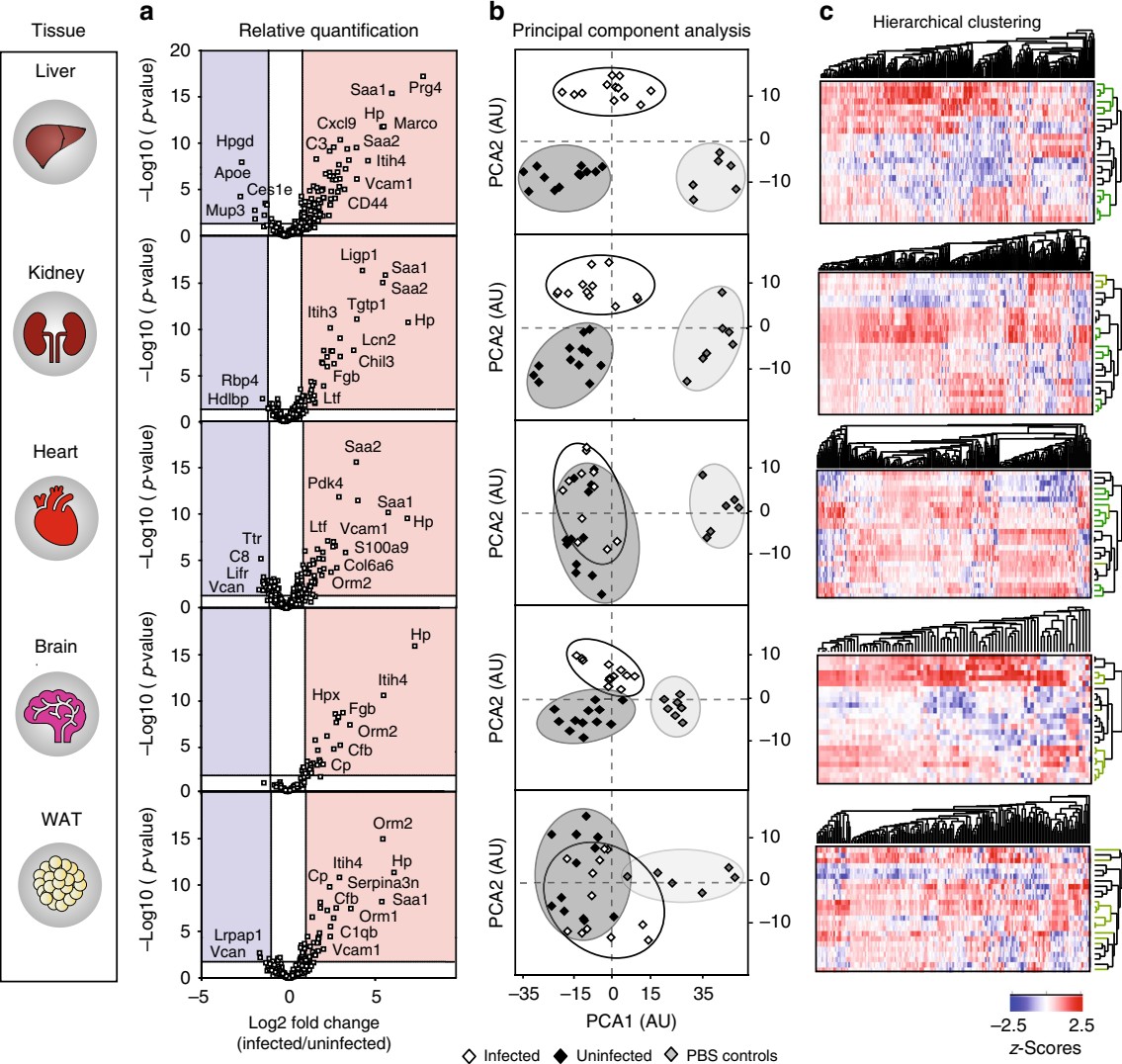

**Fig. 5** Changes in the vascular cell surface proteome in a murine model of MRSA-sepsis. Biotinylation perfusions were coupled to an automated shotgun proteomics workflow to identify organ-specific vascular targets changing in a murine model of MRSA-sepsis. We performed 4 separate experiments, which resulted in the simultaneous proteomics profiling of infected ($n = 12$) and uninfected ($n = 12$) mice across 5 major organs (liver, kidney, heart, brain, and WAT). Differential expression analysis of proteins significantly changing during infection showed that the examined organs displayed a clear hierarchy in terms of the type and amounts of vascular proteome that was altered during sepsis, with liver samples being among the most severely affected. Principal component analysis (PCA) of the identified proteins segregated the liver, kidney and brain tissues into infected and uninfected groups, but was less specific for WAT and heart stratification (**b**). Unsupervised hierarchical clustering of the data revealed dramatic proteome changes at 24 h post-infection with large protein clusters being differentially regulated across all examined organs (**c**)

kidney, and heart, many proteins were altered between infected and non-infected samples, whereas in brain and WAT only a few proteins showed significant changes (Fig. 5a). The liver pattern stood clearly out from the other organs. Also, some of the hepatic proteins that underwent dramatic changes in labeling included lubricin/proteoglycan 4 (Prg4), macrophage receptor with collagenous structure (Marco), C-X-C motif chemokine ligand 9 (Cxcl9), as well as others. We subjected the data to principal component analysis (PCA) to determine if the organ protein patterns combined with their LFQ-intensities could segregate the data into meaningful groups. Liver samples were clearly separated by PCA analysis into infected and non-infected clusters (Fig. 5b). Kidney and brain also stratified in a similar fashion. On the other hand, PCA analysis of WAT and heart samples did not separate the infected and uninfected groups from each other. Unsupervised hierarchical clustering of the proteomics data revealed complex expression patterns. Heat maps generated to visualize

the clustered data showed that all tissues had large clusters of proteins enriched or depleted by infection (Fig. 5c).

A shared group of 57 proteins was consistently found to undergo changes in all examined organs, whereas other proteins were changing in a tissue-specific manner (liver: 88, kidney: 75, heart: 85, brain: 8, and WAT: 18) (Fig. 6a and Supplementary Data 3). Functional enrichment analysis of the shared proteome retrieved biological pathways associated with coagulation, complement, ECM-remodeling, and *Staphylococcus aureus* infection (Fig. 6b). This proteome signature consisted of acute phase reactants (e.g. Hp, Saa2), members of the alternative pathway of the complement cascade, for example complement factor b (Cfb), Cfh, and the matrix proteins basal cell adhesion molecule (Bcam) and nidogen 1 (Nid1). Interestingly, some of these shared proteins experienced very large fold-changes in all organs (e.g. fold-change of haptoglobin (HP) >100 in all tissues) whereas other proteins had a more scattered expression, with large fold-

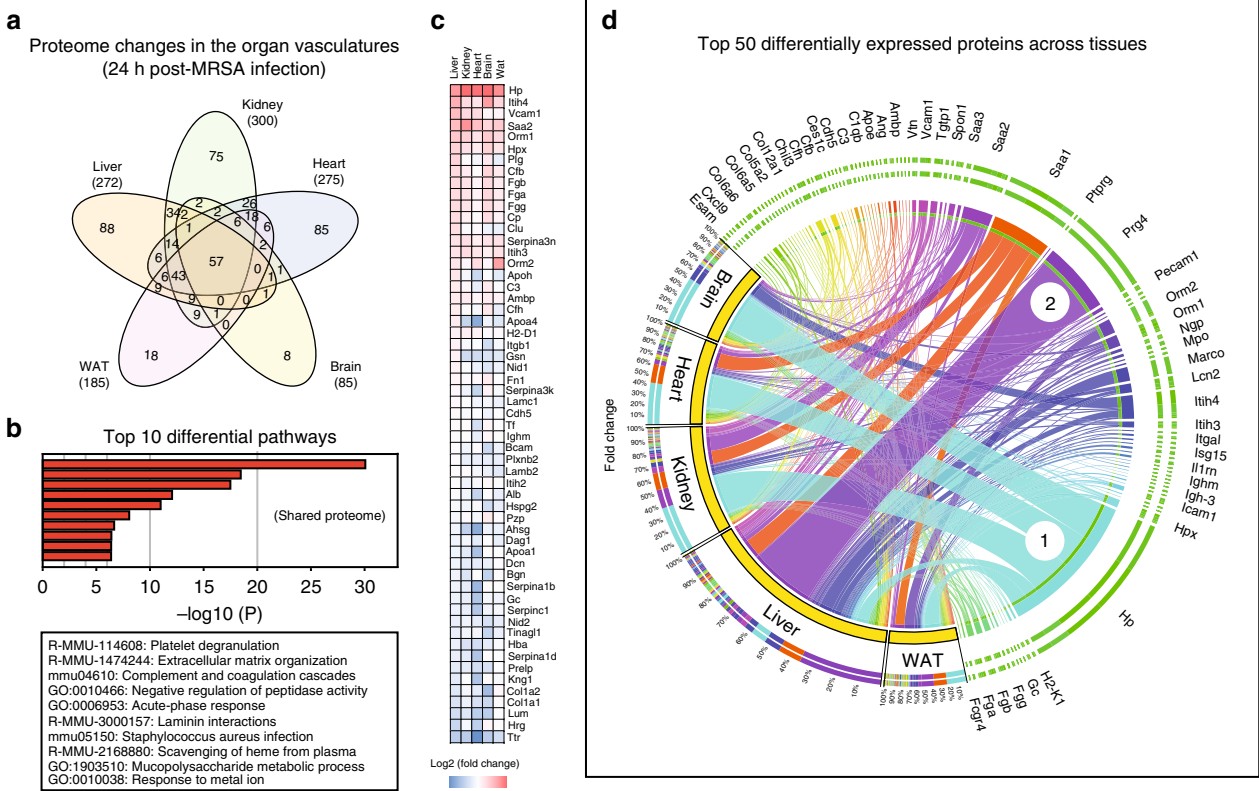

**Fig. 6** Remodeling of the vascular surfaces during MRSA-sepsis. Venn diagrams of significant proteins hits across the organs (infected $n = 12$, uninfected $n = 12$) revealed that a total of 57 proteins were shared among all examined tissues, whereas other targets were changing in a tissue-specific fashion (**a**). Functional enrichment analysis using the Metascape tool indicate that biological processes related to coagulation, acute phase responses and ion hemostasis are highly enriched in the shared proteome across all organs (**b**). Heat map of the organ average fold-change values for all proteins in the shared category (**c**). Circos plot depicting the normalized fold-changes of the top 50 differential proteins across five organs (**d**). Each protein value is expressed as a ribbon with a unique color, the width of which corresponds to the normalized fold-change of that protein as a percentage of the summed fold-changes of all identified proteins in each tissue. Haptoglobin (Hp) is marked with an encircled number 1 to illustrate proteins displaying large induction in all tissues, whereas proteoglycan 4 (Prg4) is marked with an encircled number 2 to highlight proteins displaying very large fold-changes in a tissue-specific fashion

changes in one organ and modest induction in the others (e.g. >30-fold change of alpha-1-acid glycoprotein 2 (Orm2) in WAT, whereas ~2–6-fold in all other tissues) (Fig. 6c).

The fold-changes associated with the top 50 differential proteins in each tissue were ranked and visualized as Circos plots (Fig. 6d). In this representation, each protein corresponds to one ribbon, and the width of the ribbon indicates the normalized fold-change of an individual protein as a percentage of the summed fold-changes of all identified proteins in a particular organ. Strong induction of Hp was observed in all tissues accounting for 7% of all proteome changes in the liver, 36% in kidney, 34% in heart, 45% in brain, and 23% in WAT (Fig. 6d). This is clearly shown in the Hp ribbon, which further divides into 5 sub-ribbons that connects back to the protein distribution of each organ. Interestingly, the pattern of the liver was unique because it was largely dominated by liver-specific markers, as opposed to the other organs where the largest fold-changes were instead associated with the shared vascular proteome. Proteoglycan 4 (Prg4) accounted for 34% of all liver vascular surface proteome alterations but remained undetected in the other tissues, except for the heart where it corresponded to 1% of the cardiac proteome changes.

We identified other tissue-specific markers, which were further subjected to pathway enrichment analysis to delineate organ-specific pathologies (Supplementary Fig. 6a–e and Supplementary Data 3). In the liver, neutrophil degranulation was the most

enriched biological process, reflecting the presence of multiple neutrophil-derived proteins such as myeloperoxidase (MPO), an essential factor for neutrophil antimicrobial responses[30,31]. Other liver-specific pathways were related to cell-adhesion, integrin signaling, and glycosaminoglycan metabolism, especially hyaluronan/hyaluronic acid (HA). Kidney samples were enriched in multiple renal adhesion proteins such as the kidney-specific cadherin-16, nephronectin, and nephrin, as well as in markers of renal epithelium morphogenesis (Supplementary Fig. 6b). Similarly, heart-specific processes were highly enriched in proteins related to muscle contraction and markers of hypertrophic cardiomyopathy (HCM), reflecting potential ongoing heart failure (Supplementary Fig. 6c). Finally, a smaller number of organotypic proteins and hence a smaller number of differential pathways were detected in brain and WAT tissues, compared with liver, kidney and heart (Supplementary Fig. 6d, e).

**Changes in hyaluronan turnover and recognition**. As mentioned above, some liver-specific markers could be grouped based on their ability to influence HA recognition and turnover. HA is a megadalton acidic polysaccharide involved in many immunomodulatory processes, including recruitment of activated neutrophils to the liver during inflammatory responses[32]. Among the proteins enriched in HA metabolism and function, we found CD44, one of the main HA receptors that regulates leukocyte

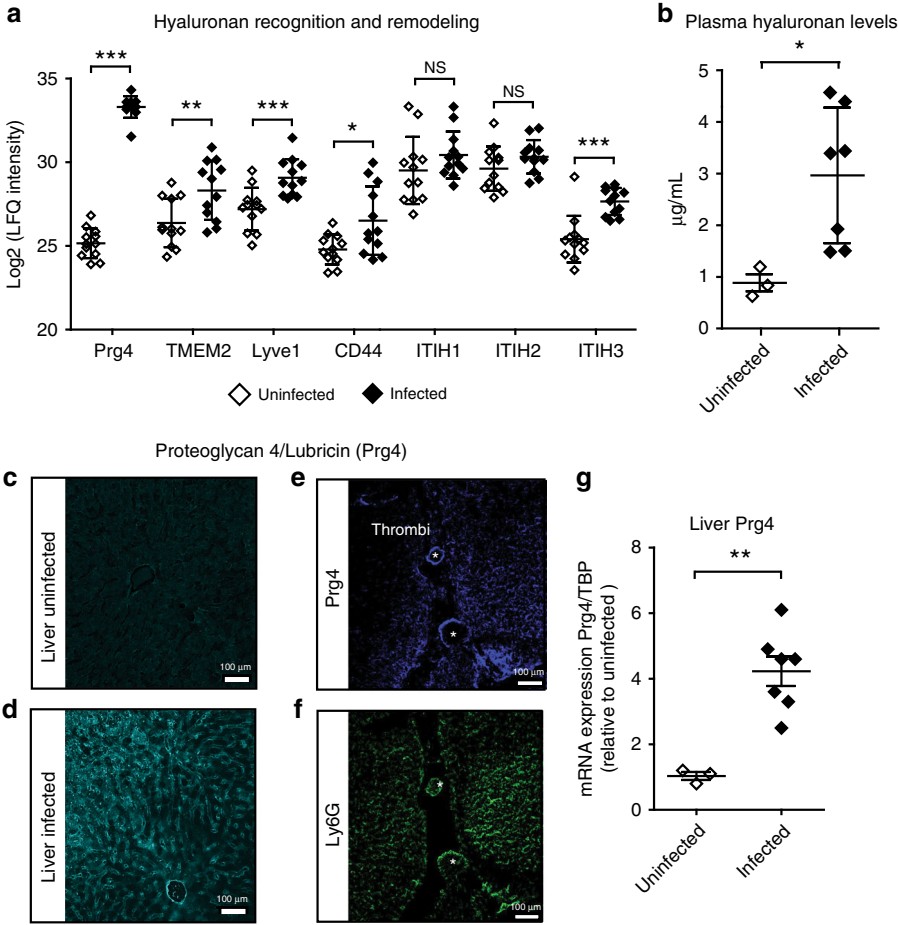

**Fig. 7** Changes in hyaluronan and hepatic hyaluronan-binding proteins. Relative label-free quantification (LFQ) analysis of proteomic changes in the hepatic vasculature during sepsis reveals differential abundance of multiple targets involved in hyaluronic acid recognition and turnover (**a**). The levels of circulating hyaluronic acid in plasma at 24 h post-infection were also significantly increased in a separate cohort of MRSA-infected animals ($n = 7$) compared with controls ($n = 3$) (**b**). MRSA infection increases expression and deposition of Prg4 along the central veins and the sinusoidal microvasculature (**c**, **d**). Prg4-immunoreactivity was also found at the edges of large necrotic thrombi in association with Ly6G + neutrophils (**e**, **f**). qPCR analysis in a separate cohort of mice (infected livers $n = 7$, uninfected controls $n = 3$) demonstrated increased expression of hepatic Prg4-mRNA levels already at 12 h post-infection (**g**). Data are represented as mean ± SD. ***$p < 0.001$, **$p < 0.01$ and *$p < 0.05$ by two-sided Student's $t$-test. Scale bar, 100 μm

trafficking and extravasation; the lymphatic vessel endothelial hyaluronan receptor 1 (Lyve1), the primary HA receptor in lymphatic endothelium; Tmem2, an endothelial cell surface hyaluronidase; and Prg4, a lubricating and anti-inflammatory competitor for HA binding to CD44 (Fig. 7a)[33]. Notably, endothelial HA binds only weakly to CD44, unless the former becomes cross-linked to one or more of the inter-alpha-trypsin inhibitor heavy chains[32]. Cross-linked HA constitutes a high-affinity ligand for neutrophil CD44 during leukocyte extravasation in the liver sinusoids. However, the identity of the heavy chains involved in this modification remains unknown. At least 6 closely related heavy chains are expressed by many cell types, but only Itih1, Itih2, and Itih3 can be cross-linked to HA[34]. Interestingly, we detected these three heavy chains in both normal and septic livers, but no significant differences were observed regarding the levels of Itih1 and Itih2 during infection. In contrast, a 5-fold increase in Itih3 was consistently detected in liver samples, suggesting a potential role in HA-maturation during liver inflammation. Moreover, increased HA plasma levels has been identified as a sensitive marker of glycocalyx remodeling and liver dysfunction during sepsis[35,36]. Indeed, at 24 h post-infection, plasma HA levels were elevated more than 6-fold in MRSA-infected samples compared to controls (Fig. 7b and Source Data File).

We focused on Prg4, since this HA-binding protein displayed the largest induction of all detected liver proteins, in all liver replicates, and across all independent experiments (Supplementary Data 2). Also known as lubricin, Prg4 is a secreted high molecular weight glycoprotein composed of globular domains connected by an extended mucin region. Prg4 is abundantly expressed in synovial fluid and its glycosylation plays an important role in the lubrication of articular joints. Although extra-articular expression of Prg4 is known to occur, its physiological role or involvement in liver disease has never been reported[37].

Expression and localization studies of Prg4 by immunofluorescence microscopy showed dramatic increase and deposition of Prg4 in the microvasculature of septic livers (Fig. 7c, d) compared with uninfected control liver. Strong Prg4-immunoreactivity was detected both in vascular and perivascular regions, in agreement with the results from the biotinylation experiments. Of note, Prg4-staining was also detected in the periphery of large thrombi in the vicinity of necrotic areas, and in association with ly6G-positive neutrophils (Fig. 7e, f). Finally, quantitative PCR analysis of liver tissue at 12 h post-infection revealed a 4.2-fold increase in *PRG4* mRNA compared to uninfected controls (Fig. 7g and Source Data File). Taken together, these signatures suggest that

remodeling of HA and HA-modifying proteins is a liver-specific phenomenon activated during MRSA sepsis.

## Discussion

Remodeling of the vascular glycocalyx is a hallmark of devastating diseases such as atherosclerosis, cancer and dysregulated inflammatory responses. Both glycocalyx shedding and repair (remodeling) are tightly regulated by a variety of mechanisms that can be triggered in response to physiological and/or pathological stimuli. A mechanistic understanding of these processes has been hampered by a general lack of tools to properly address their molecular basis. Prior studies have demonstrated the use of systemic perfusion to tag murine vascular surfaces in vivo, using ester derivatives of biotin[21,22]. In principle, this delivery route creates a window to track molecular events at the blood/tissue interface and to gain insights into changes in the vascular proteome in a broad spectrum of diseases. However, these previous studies were based on labor-intensive protocols, with limited reproducibility and throughput. Additionally, chemical labeling has mainly been interfaced with off-line chromatographic separations and low-resolution MS. Nevertheless, these techniques have potential as generic tools to interrogate preclinical models of vascular diseases such as sepsis, where multiple host tissues are simultaneously engaged.

In the present study, we combined systemic biotinylation of murine tissues with an automated proteomics workflow to profile the vascular surfaces in a more robust and quantitative fashion. Indeed, multidimensional chromatography and high-resolution proteomics profiling of murine organs confirmed the presence of a complex protein landscape embedded in the vascular glycocalyx. In keeping with previous reports, proteins accessible to the vascular flow comprised a large number of adhesion molecules (including many classical endothelial markers), receptor tyrosine kinases, enzymes, scavenger receptors, proteoglycans, and plasma proteins. Notably, we observed the presence of a shared core proteome, probably essential to basic vascular functions of most tissues, along with concomitant organ-specific vascular signatures. Phenotypic heterogeneity of the vasculature is a phenomenon that has become widely recognized[38,39]. Organotypic specification of vascular tissues allows the organs to meet basic requirements of oxygenation, nutrient acquisition, hemodynamic regulation, and immunosurveillance, across a wide range of local environments. Organs such as the liver or kidney are specifically involved in regulating the rate of exchange and/or clearance of ions, metabolites, proteins, and bacterial toxins from circulation, and thus some organ-specific functions need to be wired at the vascular level. Consistent with this idea, we found proteomic signatures specific to hepatic or renal vascular surfaces that reflected specific functions of those organs. For example, in the liver multiple proteins involved in lipid clearance were detected together with a large repertoire of scavenger receptors. In contrast, kidney displayed proteome signatures consistent with its role in controlling fluid balance and blood pressure. Recently, in vivo phage display assays facilitated identification of peptides capable of homing to specific vascular beds in vivo[40]. These approaches have been useful in generating tools for targeted delivery of compounds to specific vascular "zip codes", and they highlight the tremendous organotypic heterogeneity of the vascular surfaces. Similarly, recent developments in single cell sequencing have facilitated detailed profiling of the brain vasculature, revealing specific molecular signatures associated with zonation along the arteriovenous axis[41].

To obtain a holistic picture of vascular changes taking place during sepsis, we adapted a murine model of MRSA-sepsis and subjected the animals to chemical labeling and a proteomics workflow, selecting several organs for molecular profiling. We constructed a label-free semi-quantitative proteomics atlas of the murine vascular cell surface proteome at the organ level (liver, kidney, heart, brain, and WAT), and demonstrated that a septic insult triggered by MRSA bacteremia causes substantial remodeling of the vascular surfaces. More specifically, we demonstrated that MRSA-sepsis results in enrichment of acute-phase reactants and specific matrix proteins in the vasculature of multiple tissues. Interestingly, some of the identified proteins, such as Hp and the serum amyloid proteins Saa1 and Saa2, have already demonstrated predictive value for diagnosis of systemic inflammatory processes, including neonatal sepsis[42]. We also identified alterations of an array of proteins and biological processes in particular tissues, including potentially novel pathways associated with hepatic HA-recognition and turnover. Our proteomics screening singled out the liver as a hotspot for MRSA sepsis pathology, a pattern further confirmed by blood chemistry and histopathological evidence of liver dysfunction, necrosis and thrombosis. The levels of many proteins in the hepatic vasculature changed during infection. One in particular, Prg4, consistently dominated the global pattern of proteome changes in the liver. Prg4 can oligomerize and deposit in the superficial zone of the articular cartilage, acting as a lubricating gel to reduce friction[43]. Due to the large amount of negatively charged carbohydrates decorating its mucin domain, Prg4 can impact the adhesion properties of synovial cell populations, bacteria and immune cells[44]. Interestingly, Pgr4 is also expressed in the liver but a role in basic liver physiology or pathology has never been described. Here we report that PRG4 gene expression is rapidly induced during MRSA bacteremia, resulting in its secretion and abundant vascular deposition in the liver. Notably, Prg4 was also detected in association with the surface of neutrophil populations, adhering to the edges of large necrotizing thrombi, partially occluding the perfusion of the liver. It is conceivable that vascular deposition of Prg4 during liver inflammation might impact the adhesion of bacteria to the glycocalyx and influence how immune cells such as neutrophils interact with the activated endothelium. A mechanistic understanding of the role of Prg4 during acute hepatic injury, and especially during MRSA sepsis, warrants future studies.

Circulating glycocalyx fragments are most likely good markers of organ-specific pathologies. However, we do not know if some of the proteome alterations reported in this study can also be detected in plasma. Plasma proteomics has been widely applied to identify fluid biomarkers of sepsis[28,45,46]. One major challenge with these analyses though is the broad dynamic range of plasma proteins, which exceeds 10 orders of magnitude[47]. Profiling molecular changes in the vascular compartment can aid in the search for novel plasma markers by defining molecular targets undergoing remodeling at the vascular wall. Having pre-knowledge of what kind of candidate molecules to seek might facilitate the application of targeted proteomic assays to the plasma proteome to increase sensitivity and coverage. Indeed, future studies will be directed to the validation of promising candidates in plasma samples to assess their predictive value. Finally, this study focused on a single time-point, but the dynamic nature of the sepsis response can only be fully addressed with temporal-spatial resolution. Tracking some of these markers over time will be imperative to achieve that final goal.

## Methods

**Bacterial strain and preparation**. Methicillin-resistant *Staphylococcus aureus* (MRSA USA300 TCH1516) was originally isolated from an outbreak in Houston, Texas and caused severe invasive disease in adolescents[48]. MRSA was routinely grown at 37 °C on Todd-Hewitt agar (Difco) or in liquid cultures of Todd-Hewitt broth (THB, Difco) with agitation (200 rpm). Bacteria were inoculated into 5 mL of

fresh THB and incubated overnight. 400 μL of overnight culture was inoculated into 6 mL of fresh THB and incubated to $OD_{600} = 0.4$. Bacteria were centrifuged, washed twice with PBS, and suspended in PBS at $5 \times 10^8$ cfu/mL.

**Animal experiments.** For MRSA infection, 8–10-week-old C57Bl/6 male and female mice were injected i.v. through the retroorbital sinus with 100 μL PBS as a control group or with $5 \times 10^7$ cfu (100 μL) MRSA. At 24 h post-infection, animals were euthanized by isoflurane and immediately processed using systemic chemical perfusions. CFU in the MRSA inoculum were enumerated by serial dilution on Todd Hewitt Agar plates to ensure consistent CFU dosing across experiments. Animals were housed and bred in Individual Ventilated Cages in a Specific Pathogen Free background, in vivaria approved by the Association for Assessment and Accreditation of Laboratory Animal Care located in the School of Medicine, UC San Diego. All experiments were performed in accordance with relevant guidelines and regulations following standards and procedures approved by the UC San Diego Institutional Animal Care and Use Committee (protocol #S99127 and #S00227M).

**Systemic chemical perfusions.** In vivo biotinylation was essentially conducted as reported[21] with some minor changes. Briefly, animals were anesthetized using isoflurane in a closed chamber and a median sternotomy was performed. The left ventricle of the heart was punctured with a 25-gauge butterfly needle (BD Vacutainer) and a small cut was made in the right atrium to allow draining of perfusion solutions. All perfusion reagents were ice-cold and were infused using a perfusion pump (Fischer scientific). Blood components were quickly washed out with PBS for 5 min at a rate of 5 mL/min. A solution containing 100 mM EZ-link Sulfo-NHS-biotin (Thermo Fischer) in PBS, pH 7.4 was used to perfuse the animals at 3 ml/min for 10 min. Finally, animals were perfused with the quenching solution (50 mM Tris-HCl, pH 7.4) at 3 ml/min for 5 min. Control animals were perfused in exactly the same way but with PBS.

**Organ preparations.** Mouse organs were harvested and homogenized using zirconia/silica beads, (1 mm diameter, Biospec) in a benchtop MagNA Lyser instrument (Roche). Homogenization buffer contained 5 M urea, 0.25 M NaCl and 0.1% SDS. Samples were briefly centrifuged at $16100 \times g$ for 5 min to sediment insoluble tissue debris. The clear supernatant was transferred to a new tube and protein was quantified by BCA assay (Thermo Scientific) as per manufacturer instructions and stored at −80 °C until further analysis.

**SDS–PAGE and in gel western blot.** Organ homogenates were resolved by SDS–PAGE (Bis-tris 4–12%). After electrophoresis, gels were incubated with 50% isopropanol + 5% acetic acid for 15 min, followed by a 15 min wash with ultrapure water. Gels were further incubated with streptavidin-IRDye680 (LI-COR Biosciences) for 1 h with gentle shaking in the dark. Gels were washed three times for 10 min in PBS + 0.1% Tween 20. After a last wash with PBS for 5 min, the gel was imaged using an Odyssey Infrared Imaging System (LI-COR Biosciences).

**Histological analysis and immunofluorescence.** Tissues were fixed in 10% buffered formalin (Fischer Chemical) for 24 h, followed by submersion in 70% ethanol for at least 24 h. The samples were paraffin embedded and sectioned (3 μm), and stained with hematoxylin/eosin. For immunofluorescence of perfused tissues, organs were harvested immediately following systemic Sulfo-NHS-Biotin perfusion and fixed in ice-cold PBS + 4% paraformaldehyde for 18–24 h with gentle end-over-end agitation. Fixed organs were placed in 40% sucrose solution overnight. Saturated organs were then submerged in Optimal Cutting Temperature compound (OCT) (Sakura) and flash frozen in cassettes submerged in 2-methylbutane chilled with dry ice. Sections (20 μm) were permeabilized and stained for 1 h with 1 μg/mL Streptavidin Alexa Fluor 488 (Invitrogen) and 20 μg/mL Isolectin B4 Alexa Fluor 594 (Thermo Fischer Scientific), followed by mounting medium containing DAPI (Thermo Fischer Scientific). Lubricin/PRG4 Antibody (NBP1-19048, Novus Biologicals) was diluted 1/50 and detected with secondary goat anti rabbit Alexa Fluor 647 (A-21245, Invitrogen). Sections were mounted on glass slides under #1.5 coverslips. All Z-stacks were acquired with an inverted Zeiss LSM 880 confocal with FAST AiryScan, using either a ×10 Plan-Apochromat 0.45NA objective or a ×40 LD LCI Plan-Apochromat 1.2NA immersion objective as indicated in the figure legend. Images were processed using the in line AiryScan processing module in Zen Black. For intensity comparisons, all acquisition parameters were standardized to the following: Red channel 561 nm, 721 master gain, 8.7% laser power, 3.7 AiryScan parameter; Green channel 488 nm, 676 master gain, 1.2% laser power, 4.1 AiryScan parameter; DAPI channel 405 nm, 732 master gain, 3.8% laser power, 3.8 AiryScan parameter.

**qPCR analysis.** mRNA was isolated from whole liver extracts, reverse transcribed, and quantitated by qRT-PCR using the following primers for murine *PRG4*: 5′-CAG GAC AGC ACT CCA TGT AGT-3′ (reverse) and 5′-GGG TGG AAA ATA CTT CCC GTC-5′ (forward). mRNA was extracted using the RNeasy Kit (Qiagen). cDNA was synthetized using SuperScript III First-Stand Synthesis kit (Invitrogen).

Expression was quantified using the $2^{-\Delta\Delta CT}$ method and TBP was used to normalize the expression of the target genes between samples.

**Determination of hyaluronic acid concentration.** Hyaluronan levels in plasma were measured using a Hyaluronan Quantikine Enzyme-Linked Immunosorbent Assays (ELISA) kit (R&D Systems) according to the manufacturer's recommendation.

**Determination of plasma levels of ALT and AST.** Blood was collected via cardiac puncture and placed in a pro-coagulant serum tube (BD Microtainer #365967) for 4 h at room temperature. Serum was isolated by spinning the tubes at $2000 \times g$ and collecting the supernatant. All samples were frozen and thawed once before analysis. ALT/AST was measured on a Cobas 8000 automated chemistry analyzer (Roche) with a general coefficient of variance of <5%.

**Analysis of bacteria colony-forming unit counts.** Organs of interest were placed in a 2 mL tube (Sarstedt #72.693.005) containing 1 mL ice-cold PBS and 1.0 mm diameter Zirconia/Silica beads (Biospec Products #11079110z). Samples were homogenized using a MagNA Lyzer (Roche) for 2 min at 6000 rpm. An aliquot of each organ sample was serially diluted in PBS and plated on Todd-Hewitt Agar to enumerate CFU.

**Purification of biotinylated proteins.** Biotinylated proteins were purified from homogenized organs (3 mg protein) using a Bravo AssayMap platform and AssayMap streptavidin cartridges (Agilent). Briefly, cartridges were prewashed with 50 mM ammonium bicarbonate (pH 8), and then samples were loaded. Non-biotinylated proteins were removed by extensively washing the cartridges with 8 M urea in 50 mM ammonium bicarbonate buffer (pH 8). Cartridges were washed with Rapid digestion buffer (Promega, Rapid digestion buffer kit) and bound proteins were subjected to on-column digestion using mass spec grade Trypsin/Lys-C Rapid digestion enzyme (Promega, Madison, WI) at 70 °C for 2 h. Released peptides were desalted in the Bravo platform using AssayMap C18 cartridges and the organic solvent was removed by vacuum centrifugation (SpeedVac). Samples were stored in −20 °C prior to LC–MS/MS analysis.

**LC–MS/MS analysis.** Dried peptides were reconstituted with 2% acetonitrile, 0.1% formic acid, and quantified by modified BCA peptide assay (Thermo Fisher Scientific). Equal peptide amounts derived from each sample were injected and analyzed by LC-MS/MS using a Proxeon EASY nanoLC system (Thermo Fisher Scientific) coupled to a Q-Exactive Plus mass spectrometer (Thermo Fisher Scientific). Peptides were separated using an analytical $C_{18}$ Acclaim PepMap column ($0.075 \times 500$ mm, 2 μm; Thermo Scientific) equilibrated with buffer A (0.1% formic acid in water) and eluted in a 93-min linear gradient of 2–28% solvent B (100% acetonitrile) at a flow rate of 300 nL/min. The mass spectrometer was operated in positive data-dependent acquisition mode. MS1 spectra were measured with a resolution of 70,000, an automated gain control (AGC) target of 1e6 and a mass range from 350 to 1700 m/z. Up to 12 MS2 spectra per duty cycle were triggered, fragmented by higher energy collisional dissociation (HCD), and acquired with a resolution of 17,500 and an AGC target of 5e4, an isolation window of 1.6 m/z and a normalized collision energy of 25. Dynamic exclusion was enabled with duration of 20 s.

**2D-LC-MS/MS analysis.** Dried samples were reconstituted in 0.1 M ammonium formate pH ~10 and analyzed by 2D-LC–MS/MS using a 2D nanoACQUITY Ultra Performance Liquid Chromatography (UPLC) system (Waters corp., Milford, MA) coupled to a Q-Exactive Plus mass spectrometer (Thermo Fisher Scientific). Peptides were loaded onto the first-dimension column, XBridge BEH130 $C_{18}$ NanoEase (300 μm × 50 mm, 5 μm) equilibrated with solvent A (20 mM ammonium formate, pH 10, first-dimension pump) at 2 μL/min. The first fraction was eluted at 17% of solvent B (100% acetonitrile) for 4 min and transferred to the second dimension Symmetry $C_{18}$ trap column $0.180 \times 20$ mm (Waters corp., Milford, MA) using a 1:10 dilution with 99.9% second dimensional pump solvent A (0.1% formic acid in water) at 20 μL/min. Peptides were then eluted from the trap column and resolved on the analytical $C_{18}$ Acclaim PepMap column ($0.075 \times 500$ mm, 2 μm particles; Thermo Scientific) at low pH by increasing the composition of solvent B (100% acetonitrile) from 1 to 38% (non-linear) over 96 min at 300 nL/min. Subsequent fractions were generated with increasing concentrations of solvent B. The first-dimension fractions were eluted at 19.5, 22, 26, and 65% solvent B, respectively. The mass spectrometer was operated in positive data-dependent acquisition mode. MS1 spectra were measured with a resolution of 70,000, an AGC target of 1e6 and a mass range from 350 to 1700 m/z. Up to 12 MS2 spectra per duty cycle were triggered, fragmented by HCD, and acquired with a resolution of 17,500 and an AGC target of 5e4, an isolation window of 1.6 m/z and a normalized collision energy of 25. Dynamic exclusion was enabled with duration of 20 s.

**Mass spectrometry data analysis.** MS Raw.files were processed in the MaxQuant platform[49] (version 1.6.1.0) and searched by the Andromeda search engine[50] against the mouse UniProt FASTA database (downloaded 06–02–2017) and against

a common contaminant database. Search parameters were set as follows: enzyme, trypsin/LysC with up to 2 potential missed cleavages; fixed modifications, carbamidomethyl on cysteines; variable modifications, oxidation of methionine and acetylation of protein N-terminus; minimum peptide length, 7. The false discovery rate (FDR) for both peptide and protein identifications was set to 1% and was calculated by searching the MS/MS data against a reversed decoy database. Allowed mass deviation for precursor ions was set to 5 ppm and for peptide fragments was set to 20 ppm. Label-free quantifications (LFQ) was based on a minimum of 2 counts, minimum number of neighbors was set to 3 and average number of neighbors was 6. The match between runs option was applied with a match time window of 0.7 min and an alignment time window of 20 min.

**Statistical analysis**. Bioinformatics and statistical analysis of proteomics results were conducted in the Perseus statistical suite (version 1.6.5.0)[51] of the MaxQuant computational platform. MaxQuant results were imported into Perseus and identified hits were filtered based on number of peptides (>2) and number of MS/MS scans for each peptide (>2). Missing values were addressed by requiring a cut-off corresponding to 75% valid values in at least one group (infected + biotin, uninfected + biotin, or PBS controls). Remaining missing values were imputed from the normal distribution using a width of 0.3 and a down shift of 1.8. Control samples from PBS-perfused samples were used to subtract the background, keeping protein identifications in labeled samples displaying at least a 2-fold change enrichment. Statistically significant changes between groups were assessed by two-way analysis of variance (ANOVA) with a permutation-based false discovery rate (FDR) for multiple test correction and truncation after 250 randomizations. Hierarchical clustering was applied using Euclidean distances and preprocessing with k-means. The statistical significance of the blood chemistry analysis was determined using a two-tailed Student's t-test in GraphPad prism (version 5.03). Pathway and GO term analysis were conducted using the online version of Metascape and DAVID server. Louvain clustering was performed in Python 3.6.6 using the python package python-Louvain located on the Python Package Index. Clustering was done running default parameter settings.

**Reporting summary**. Further information on research design is available in the Nature Research Reporting Summary linked to this article.

## Data availability
The proteomics raw data and metadata have been deposited in the MassIVE archive PXD013513. The source data underlying Figs. 4a, c, 7b and g are provided as a Source Data file.

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

## Acknowledgements

This work was supported by grant P01HL131474 to J.D.E. and J.W.S. N.L. and J.S. also acknowledge support from NIGMS R35 GM119850. J.W.S. was supported by R01 GM107523. G.G. was supported by Microbial Sciences Graduate Research Fellowship Awards 1-F17GG and 1-F18GG. Proteomics analysis was done at the Proteomics Core Facility, Sanford-Burnham-Prebys Medical Discovery Institute, La Jolla, CA, USA.

## Author contributions

A.G.T. and J.D.E. conceived the study and designed it together with G.G. and J.W.S.; A.G.T., G.G., A.R.C. and H.C. performed all the experiments. A.G.T., G.G., J.S., V.N., N.L. and J.D.E. interpreted the data. N.V. performed the histopathological analysis. A.G.T. and J.D.E. wrote the paper with significant input from all the coauthors.

## Competing interests

The authors declare no competing interests.
