## [Peer Review File · Nature Communications]

Reviewers' comments:

Reviewer #1 (Remarks to the Author):

The manuscript describes the application of the in vivo biotinylation technique, in combination with mass spectrometry, for the characterisation of changes in the abundance of proteins of the vasculature (or closely associated with the vasculature) in a mouse model of Staph aureus sepsis.

The methodology described in the paper is not novel (has indeed been described 15 years ago), but the application to sepsis is both novel and important.

Figure 1 may be misleading, as it suggests that only luminal proteins would be modified by the reactive ester derivative of biotin. In reality, the reagent readily extravasates and other components (e.g., extracellular matrix proteins) are detected. My recommendation would be to modify the Figure, indicating that also perivascular structures may be biotinylated.

The authors write that "previous studies have suffered from low throughput due to technical limitations at the time, and quantitative strategies were never pursued". In my opinion, this statement is not correct and may have to be reformulated. Specifically:

- The quantification in previous articles on in vivo quantification by the Neri group was performed by 2D-peptide mapping, a technology which basically provides relative quantification by comparing the intensity of mass spectrometry signals with the intensity of internal standard peptides, spiked into each chromatographic fraction. It is not obvious to me (and it has not been demonstrated) that the LFQ method used in the paper may provide a more reliable quantification. It is true, however, that MS instruments have improved over the last 15 years
- Even previous methodologies allowed the confident identification and relative quantification of several hundred non-redundant proteins. See for example:

In vivo biotinylation of the vasculature in B-cell lymphoma identifies BST-2 as a target for antibody-based therapy.

Schliemann C, Roesli C, Kamada H, Borgia B, Fugmann T, Klapper W, Neri D.

Blood. 2010 Jan 21;115(3):736-44.

A proteomic approach for the identification of vascular markers of liver metastasis.

Borgia B, Roesli C, Fugmann T, Schliemann C, Cesca M, Neri D, Giavazzi R.

Cancer Res. 2010 Jan 1;70(1):309-18.

The name Rybak is misspelled

The paper is rich in information and, as mentioned above, is an important contribution to the field. From my perspective, ideally, more immunohistochemical confirmation of unregulated targets and fewer gene ontology figures would provide a benefit to the quality of the paper. Nonetheless, the clear identification of regulated proteins in Figure 4 will allow other researchers to pursue their own experimental validation, if they wish to do so

Reviewer #2 (Remarks to the Author):

Key results: Toledo et al. have combined a previously reported “in vivo” biotinylation approach with an automated workflow for subsequent sample preparation and analysis using multi-dimensional liquid chromatography and high-resolution mass spectrometry. They used this method to characterize the vascular proteome in healthy and septic mice. They observed a common and an organ-specific vascular proteome. The most outstanding protein candidates were validated by quantitative PCR and/or histology.

Writing for a broad audience: From my point of view (as an infection immunologist with basic knowledge in proteomics), the wording and writing style is not yet directed at a broad scientific audience with variable backgrounds in proteomics. The results section is very technical and the important key messages get lost in too many proteomic details. In contrast, the discussion is written in a clear and concise way and brings the main messages across easily. Using such key words (panproteome, core/common proteome, variable/organ-specific proteome) would help not only the non-proteomics readership to extract the main messages from the results section. Moreover, technical terms and specialist terms are often used without explanation. Examples: line 624: GO, line 238: LFQ-normalized intensities, line 265: unsupervised hierarchical clustering. Moreover, Figure 5 uses novel ways for graphical illustrations (circo plots) without explaining them in enough detail.

Validity: At its current stage the manuscript has several flaws that need to be addressed before publication. This concerns aspects of data reproducibility, inconsistent usage of experimental data sets, incomplete methods and figure legends (all detailed below).

Reproducibility: What I miss in this paper are clear statements on the intraassay and interassay variability, an assessment of its quality and discussion of potential impact factors during sample generation and processing. The intraassay variability (animals from the same experimental group) is described in detail (lines 234-250). However, using the term “intraassay variability” will help the reader to understand what was compared here (diff. animals or different experimental groups). Moreover, what I miss here is a statement on the quality of the intraassay variability.

Importantly, the interassay variability in terms of protein identifications is briefly mentioned in line 235, but I cannot comprehend how the mentioned 20% reproducible proteins were calculated. This is discussed in detail below. It would also be of interest to see how well the LFQ-normalized intensities (averaged per experimental group) correlate between the different experimental groups. Can the PCAs for ALL individual samples (9 infected vs 9 controls) still separate infected from non-infected animals? Such information should be added to the manuscript.

Finally, I would suggest moving the final figure (Figure 8a) to the start of the MRSA chapter (after line 233). Please show CFU data for all three experiments (ideally in the same graph), if you have them. This will illustrate the intra- and interassay variability of the infection model. Moreover, please provide disease severity scores (if available).

Inconsistent usage of experimental data sets: A major drawback of the manuscript at its current state is the inconsistent usage of experimental data sets. Throughout the paper the authors seemingly randomly switch between showing data for a “representative data set” with $n=3$ per group or pooled data for all three data sets with $n=9$. For instance, after the authors just explained that their experimental groups showed “good correlations” (line 239, Figure S4), I was surprised to see in the key figure of the whole manuscript (figure 4) only volcano plots, PCA etc. for one experimental group (3 infected vs. 3 non-infected animals). In the subsequent figures 5 and 6 it seems like the complete data set was used again.

Do the authors have good reasons to show only one experimental data set instead of all data? If not, I strongly suggest to generally show the data for all three experimental groups to increase group size to 9 and hence reduce the β error.

Missing information: Several figure legends are incomplete. Information on group sizes and whether the graphs show a representative subset or the full data set are frequently missing in the figure legends (e.g. Figure 2, 3, 4, 7, partially 8).

Moreover, the listed methods are incomplete: The following methods are missing: hyaluronic acid determination in plasma (Fig 7g), CFU determination (Figure 8a), Quantification of alanine

aminotransferase (ALT) and aspartate aminotransferase (AST) in plasma (Fig 8b), quantitative PCR for PGR4 (Fig 8i).

Finally, I suggest to include a more detailed description of the animal model. If the infection per se is highly variable, this will impact on all subsequent analyses. So if available, please provide CFU data for the animals that underwent biotin perfusion. Did all experimental animals survive the septic challenge?

Appropriate use of statistics and treatment of uncertainties: As mentioned above, a major drawback of the manuscript at its current state is the inconsistent usage of experimental data sets. When only 1 data set was used, the group size (n=3) was critically low.

Suggested improvements: The key knowledge gain from this paper is the core and organ-specific vascular proteomes in healthy and septic mice. Unfortunately, the authors describe the core and organ-specific vascular proteome of healthy mice only for liver and kidneys (figure 3, S2, lines 161ff), even though they have obtained data from noninfected control mice from five organs in the following experiment. It would be a great improvement to the manuscript to use these data to define the core and organ-specific vascular proteome under physiological conditions. In consequence, the current part comparing only liver and kidney would be replaced (subchapter starting at line 161, figure 3, S2).

References: references are correctly cited. Completeness cannot be judged by me. I suggest to add a reference on the new sepsis definition sepsis-3 (Singer et al, JAMA 2016, 315(8): 801–810) as mentioned below.

Clarity and context: The abstract is clear, but a summary sentence is missing. Introduction and conclusions are appropriate.

Parts outside the scope of my expertise: I have a basic knowledge of proteomics but cannot judge the suitability of the used LC/MS methods and affiliated data analyses and statistics.

Novelty: The in vivo biotinylation approach per se has been previously described. The novel aspect of this manuscript is to embed this technique into an automated workflow for protein sample preparation and protein analyses. This provides the science community with a valuable and innovative tool to study the vascular proteome by in-vivo labelling in animal models. The described methods will they be of strong interest to others in the community and the wider field.

Potential future applications: Whether the mouse-derived potential sepsis markers in the vascular proteome could have a predictive value in the clinical setting is unsure, since the vascular proteome is not accessible in patients. Maybe circulating glycocalyx proteins might constitute good markers of general or organ-specific pathologies if they reach sufficient levels in plasma, but this has not been addressed experimentally in this study.

Comments on the main text in chronological order:

Line 83: I suggest referring to the most recent definition of sepsis (so-called Sepsis 3.0) in the article (Singer et al, JAMA 2016, 315(8): 801–810). The key element of the new sepsis definition is multi-organ failure, which is also the focus of this work. Hence, I suggest adding the sepsis 3.0 article to ref 12 and 13.

Line 189-199: This paragraph mentions numerous proteins that were enriched in the vascular proteome of either liver or kidneys. However, no reference to any figures/tables is provided. Moreover, no mathematical/statistical approach for selecting the named proteins was mentioned. How were these candidates selected?

Afterwards, datasets were analysed “more systematically” according to the authors (protein-protein associations using STRING). In this list, some of the previously mentioned candidates are suddenly missing (e.g. renin-angiotensin system for kidneys, scavenger receptors for liver). Is this due to differences in defining protein clusters or due to technical discrepancies?

Is the upper section (lines 189-199) necessary at all, if you perform the more systematical approach anyway?

Line 234: Please provide some details here: were the absolute protein numbers similar for infected vs. non-infected mice? How many proteins did you detect in non-biotinylated control animals (i.e. background signal)?

Line 235ff: You mention that “Typically, the proteomics screening identified hundreds of proteins from each individual organ. Roughly ~20% of these protein identifications were robust and were consistently identified across all experiments.” First of all, how did you calculate these 20%? Based on the numbers provided in Figure S3 I calculate that between 7.9 and 15.2% of the proteins identified in either of the 3 experiments were detected in all three experiments. Wouldn't a percentage based on the number of common proteins/total no of proteins from ONE experiment make more sense? Based on this, 30-50% of the proteins detected within one experiment were also

found in the other two biological replicates. This rate still seems very low. Please explain what factors might influence reproducibility.

Line 283ff: What you describe here (i.e. the proteome signature shared by all organs) is in other contexts often called “core proteome” or “common proteome” (e.g. in line 292). Using such key words (pan proteome, core proteome, variable/organ-specific proteome) would help the readership to extract the main messages from the results section.

Lines 314ff: This paragraph contains an extensive description of proteins involved in angiogenesis that were upregulated in the liver. However, I miss a reference to original data (Figure, tables, supplement?) Can I find data on the mentioned proteins and their tissue-specific expression somewhere in the supplement? If not, I suggest including such a list.

Line 354: Your data do not “verify that MRSA is rapidly cleared from the blood circulation and redirected to the liver” since you did not determine blood counts. Please rephrase.

Line 499ff: Please specify the type of cage used to house the mice (IVC, open top,...). Where mice housed under SOPF or SPF conditions? Where were the BL/6 mice obtained from? Please check the *S. aureus* colonization status of your mice (US vendors provide this information in the health sheet). This is important, because colonization and hence immunological priming might impact on the disease course. Did you have to submit an application for your animal experiments to local authorities? If so, please provide the application ID.

Line 524: Were the isolated proteins immediately analysed or frozen for later analyses?

Comments on figures and figure legends:

Figure 1: What is the abundant pink roundish symbol in the upper panel (blood)? If it is of relevance to the assay, you might want to explain it in the figure legend.

Legend: The term “quenching” usually describes “any process which decreases the fluorescence intensity of a given substance”, e.g. by excited state reactions, energy transfer, complex-formation and collisional quenching. (source Wikipedia). To my understanding, what you describe here as

quenching is simply a washing step which removes the unbound biotin or Sulfo-NHS-biotin. If this is correct, please change the wording here as well as in M&M (line 516) and within figure 1.

Figure 4: It should be mentioned in the legend that this figure is based on data from one experimental group (n=3).

Figure 5a: I never came across Circos plots in this context (proteomics) before and have difficulties grasping their content. Please explain them in more detail in the figure legend. What does the colour code in the outer ring mean (left side of the circos plots)? Give an example on how to read the graph (percentage – ribbon width). Should the label for the percentage values be “fold change of each protein as a percentage of the summed fold changes for all proteins” rather than “fold change”? Simply fold change implies the “common” fold change without normalization and is hence confusing.

Why did you normalize the fold-changes at all and not display the real values? The true fold-changes are nowhere mentioned in the manuscript, but would give the reader a very good estimate on how strong the observed changes are.

A maybe antiquated suggestion: Couldn't you display the same information in a table with your top protein candidates and their fold change?

Please indicate which dataset these circos plots are based on (1 or all 3 experiments).

Figure 5b and also line 283f: Please mention which data are included here (“proteins consistently identified in each tissue”). Are these infected mice only, or are the data based on a ratio of infected vs non-infected mice?

Figure 5c: please explain what is meant by “heat maps showing the frequency for each protein...” (Line 671). Same line: “resulted in a significant change between infected and noninfected mice...”

Figure 7: Title should be changed to “proteome changes in the hepatic vasculature are associated...”. Are the animals in figure 7 a-f identical with one of the experimental groups from the previous figures? If so, why not show the data for all groups? Please mention the chosen data set in the figure legend. Moreover, why are there unequal numbers of uninfected vs. infected mice (3 vs. 7) in Fig 7g? Again, are these mice from the previous experiments or do they belong to a separate experiment?

Figure 8: I suggest moving Figure 8a to the beginning of the sepsis chapter (see comment above), because these data show the reproducibility and variability of the infection model. Are these the same mice as for the proteomics data or a new lot? Please indicate in the figure legend. If you have

more than 1 data set, please show all data! If disease severity scores were documented, they could be included as well.

Figure 8b: Do you have ALT and AST values for more than three animals? Methods for ALT and AST measurement are missing in M&M section.

Figure 8e-f: I suggest to separate e&f from g&h. Please add PRG4 as image title to e&f. In addition, could you generate an overlay of g and h?

Figure S3: Please explain in the figure legend what exactly is shown here. Are these the identified proteins for the infected animals or for the non-infected controls? How do you calculate the no. of proteins from three animals (within one experimental group)?

Figure S4: You show correlation data for liver. Do you get similar results for kidneys? Please briefly mention this (without necessarily showing the data) in the results section (line 234ff).

Figure S5: I suggest to specify your groups infected, noninfected and PBS control as Infected + biotinylation; noninfected + biotinylation, uninfected w/o biotinylation.

Suppl table 2: needs a data dictionary and a format that allows reading the headings (i.e Excel). Values within one column should be displayed with the same decimal place.

Minor comments:

Line 73: change native immunity to innate immunity

Line 76: change disease associated molecular patterns to damage associated molecular patterns

Line 97: You might change the sentence to “Proteomic approaches have demonstrated changes in the glycoalyx...” to stress that the cited papers used proteomics.

Line 113/114: It would make sense to point out that the mentioned new techniques allow for semiquantitative analyses.

Line 188/189: check formatting

Line 228/229: The description of the animal groups is confusing. Please add: In addition, one non-infected and one infected mouse were perfused... This will make it much easier to understand.

Line 269: de-enriched = depleted/reduced?

Line 269: The pattern obtained

Line 514, 515, 517: "l" for litre should be "L"

Line 554: 3 mg or protein?

Line 614, 623: please provide the software versions

Figure 2: image headings: Isolectin B4 instead of IsolectinB4?

Line 654: You might mention here that the hypergeometric enrichment test looks for subcellular location of proteins.

Line 656: change your last "sentence" into a full sentence.

Figure 3d-e: inconsistent use of capital letters

Figure S1 legend: line 4: infrared-dye ☐ infrared dye

Reviewer #3 (Remarks to the Author):

The manuscript by Toledo et al describes a chemical labelling and quantitative mass spectrometry strategy to monitor the dynamic changes of the vascular cell-surface proteome during sepsis. The results indicate that the analyzed organ vasculatures undergo differential remodeling in sepsis. So far, proteome-wide analysis of changes to vascular surfaces under in vivo like conditions during sepsis has remained elusive, partly due to technical limitations. The work by Toledo et al provides an elegant solution to this problem by specifically quantifying an enriched fraction of the highly relevant vascular sub-proteome to monitor organ-specific changes during sepsis and thereby provide insights into the molecular mechanisms of multiple organ failure, a hallmark for sepsis. It can be expected that increased knowledge of the remodeling of the vascular surfaces will contribute to organ-specific changes during sepsis that can in future work facilitate the discovery of markers for the molecular classification of sepsis subtypes. Although, the work presented in this manuscript is impressive, of high quality and of high relevance for the field the following comments should be addressed.

The analysis of the quantitative differences between infected and non-infected animals (Figure 4-6) is at times difficult to follow. It is unclear why only a representative experiment is shown in Figure 4

and why different protein filtration strategies was used for Figure 5 and 6. I suspect that the reason is high variability between the biological experiments? If I understand the text correctly, the authors performed three separate biological experiments with three animals per group and experiment, which would give in total of 9 organs per organ-type. I suggest that the author reanalyze their data by merging the biological replicates into one large data matrix (9 infected/non-infected samples per organ type) and use the combined data set to i) display the number of identified proteins per organ and how frequently a given protein was detected across every organ, ii) use the data from the non-infected animals to objective classify the proteins into shared and organ-restricted proteins iii) show a heat map of the combined data or part of the data to demonstrate the technical reproducibility between experiments and iv) plot the variance distribution of the identified proteins. One of the strong points of the manuscript is the methodological developments and application of the method to sepsis. However, it is important that readers can judge the performance of the established method and that these results are adequately discussed in the discussion section.

Furthermore, it seems more straight forward to use the merged data set with the three biological experiments to calculate statistically differentially changed proteins (p-value corrected for all comparisons) between infected and non-infected animals and display this data in Figure 4 rather than showing the data from a representative experiment. If this is not possible, the authors should at least provide the results from the two other biological experiments in the supplement material in a similar manner as shown in Figure 4. In this context it would be interesting to know which of the significantly induced or repressed proteins are shared between the organs or uniquely confined to one or a few organs. The authors should also show the pca plot and heat maps for the combined data set.

In Figure 3a the authors show TIC graphs of the high-pH peptide fractions. These graphs are not very informative and could be moved to the supplement material. It is unclear why only one replicate was used in this analysis?

In Figure 5 the authors show Circos plots which are a bit difficult to interpret. I suggest that these plots are made larger. In figure 5C the authors demonstrate significant differences for 32 target proteins that were shared among all organs. I suggest that the authors provide additional fold-change plots or a heat maps with log₂ protein intensities or z-scored protein intensities for these proteins across all the analyzed organs.

Figure 7: are these samples from a representative experiment? If so why are the number of replicates different for the plasma HA analysis (Fig 7g)? Please state in the Figure legend if the p-values were corrected for multiple hypothesis testing. The authors state on line 336 that the analytical constraints were relaxed. The authors should clarify what this exactly means.

The results from the representative experiment in Figure 4 are convincing. It is however surprising that the authors do not detect increased degradation and shedding of the glycocalyx. In contrast, many proteins seem to be more abundant in the infected animals. Some of the increased proteins are abundant plasma proteins that potentially stick to the vascular cell-surfaces, but for other proteins the reason for the increased abundance is less clear. The manuscript would benefit for an extended discussion where reasons behind these results are discussed in more detail. In particular the absence of visible signs of increased degradation and shedding.

Minor comments

- Several of the figure legends are too brief. In Figure legend 4 the authors should add information stating which part of the data that was included in the analysis and provide information of the p-value correction test. In Figure legend five the authors should add text how the proteins were selected for the Circos plots.
- Supplementary figure 6d seems have different magnifications between infected and non-infected animals.
- Figure 8a, for the sake of consistency, Fat in the y-axis should be changed to WAT.
- Line 279 – SSA 2 should be changed to SAA2 to refer to serum amyloid A protein 2.
- Inter-alpha trypsin inhibitor heavy chain 3 and 4 should be designated as Itih3 and Itih4 in mice.
- Prg4 or PRG4 should be used consistently to define protein or gene names in the text.
- Add method section for qPCR
- Figure 8c and d: Add a representative image from healthy tissue as comparison

Reviewer #4 (Remarks to the Author):

The authors present a surface proteomic atlas of mouse vasculature and how this changes after infection in the context of sepsis. The authors examined several organs and show remarkable depth in their sensitivity of protein detection using the latest spectrophotometry methods. This is a method that may have applicability to a wide audience. Two fundamental elements diminish enthusiasm for the study. The main concern is the reproducibility of the technique. The proteomics only saw 20% of the protein identifications as being robust and consistent across all experiments, as

stated in the text. As an example the venn diagram of liver shows that while there were 1,472 proteins identified only 215 (15%) were consistent across the 3 mice, which show a pretty high lack of reproducibility. In practical terms, this would require a large number of mice and resources, in the way of proteomics, to gain a firm idea on what changes are likely to be real. The second major point is the sepsis information generate is largely descriptive, hence the paper could be constructed as more of a resource style article.

Minor points

The data in figure 5a, might be visualized differently. The different magnitudes of the proteins are not clearly evident in this type of plot.

Figure 8 could include a more zoomed out image to show the pathology overall.

Reviewers' comments:

We want to thank all the reviewers for their impartial reviews and for their valuable and constructive input. The manuscript has now been substantially updated according to their suggestions, as detailed below:

Reviewer #1 (Remarks to the Author):

The manuscript describes the application of the in vivo biotinylation technique, in combination with mass spectrometry, for the characterization of changes in the abundance of proteins of the vasculature (or closely associated with the vasculature) in a mouse model of Staph aureus sepsis.

The methodology described in the paper is not novel (has indeed been described 15 years ago), but the application to sepsis is both novel and important.

Figure 1 may be misleading, as it suggests that only luminal proteins would be modified by the reactive ester derivative of biotin. In reality, the reagent readily extravasates and other components (e.g., extracellular matrix proteins) are detected. My recommendation would be to modify the Figure, indicating that also perivascular structures may be biotinylated.

Response: Fig. 1 has now been changed as suggested. A key has been added where perivascular structures as well as plasma proteins associated with the vascular surfaces are also indicated as targets for biotinylation. In addition, in lines 157-159 we wrote: "strong streptavidin reactivity was also detected at the basement membrane, and within the adjacent extracellular matrix (ECM) of the endothelial cells"

The authors write that "previous studies have suffered from low throughput due to technical limitations at the time, and quantitative strategies were never pursued". In my opinion, this statement is not correct and may have to be reformulated. Specifically:

- The quantification in previous articles on in vivo quantification by the Neri group was performed by 2D-peptide mapping, a technology which basically provides relative quantification by comparing the intensity of mass spectrometry signals with the intensity of internal standard peptides, spiked into each chromatographic fraction. It is not obvious to me (and it has not been demonstrated) that the LFQ method used in the paper may provide a more reliable quantification. It is true, however, that MS instruments have improved over the last 15 years
- Even previous methodologies allowed the confident identification and relative quantification of several hundred non-redundant proteins. See for example:

In vivo biotinylation of the vasculature in B-cell lymphoma identifies BST-2 as a target for antibody-based therapy. Schliemann C, Roesli C, Kamada H, Borgia B, Fugmann T, Klapper W, Neri D. Blood. 2010 Jan 21;115(3):736-44.

A proteomic approach for the identification of vascular markers of liver metastasis. Borgia B, Roesli C, Fugmann T, Schliemann C, Cesca M, Neri D, Giavazzi R. Cancer Res. 2010 Jan 1;70(1):309-18.

Response: We agree and have changed the text to:

"Studies have demonstrated the usefulness of these approaches to profile and quantify vascular antigens in metastatic livers and B-cell lymphomas, using time-of-flight (TOF) mass spectrometry technology [24, 25]. However, new generation mass spectrometers with orbitrap-based technology are now widely available, showing increased sensitivity, scan speed and mass accuracy compared to their predecessors. These instruments facilitate high-resolution measurements of fragment ions,

improved proteome coverage, lower false discovery rates, and more robust absolute and semiquantitative proteome analysis.”

In addition, the suggested references have now been added.

The name Rybak is misspelled.

Response: Corrected

The paper is rich in information and, as mentioned above, is an important contribution to the field. From my perspective, ideally, more immunohistochemical confirmation of unregulated targets and fewer gene ontology figures would provide a benefit to the quality of the paper. Nonetheless, the clear identification of regulated proteins in Figure 4 will allow other researchers to pursue their own experimental validation, if they wish to do so

Reviewer #2 (Remarks to the Author):

Key results: Toledo et al. have combined a previously reported “in vivo” biotinylation approach with an automated workflow for subsequent sample preparation and analysis using multi-dimensional liquid chromatography and high-resolution mass spectrometry. They used this method to characterize the vascular proteome in healthy and septic mice. They observed a common and an organ-specific vascular proteome. The most outstanding protein candidates were validated by quantitative PCR and/or histology.

Writing for a broad audience: From my point of view (as an infection immunologist with basic knowledge in proteomics), the wording and writing style is not yet directed at a broad scientific audience with variable backgrounds in proteomics. The results section is very technical and the important key messages get lost in too many proteomic details. In contrast, the discussion is written in a clear and concise way and brings the main messages across easily. Using such key words (panproteome, core/common proteome, variable/organ-specific proteome) would help not only the non-proteomics readership to extract the main messages from the results section. Moreover, technical terms and specialist terms are often used without explanation.

Response: The text and graphical material have now been significantly altered to make them more understandable for a broader audience, in line with the helpful suggestions from the reviewers.

Examples: line 624: GO, line 238: LFQ-normalized intensities, line 265: unsupervised hierarchical clustering.

Response: The terms are now spelled out in the text.

Moreover, Figure 5 uses novel ways for graphical illustrations (circos plots) without explaining them in enough detail.

Response: In order to simply the figure (which is now Fig. 6 in the revised paper), we integrated all individual circos plots into one single plot that summarizes the most significant protein hits and how they distribute across the tissues. Additionally, we added the following explanatory paragraph to the main text body:

“The fold-changes associated with the top 50 differential proteins in each tissue were ranked and visualized as Circos plots (Fig. 6d). In this representation, each protein corresponds to one ribbon, and the width of the ribbon indicates the normalized fold-change of an individual protein as a

percentage of the summed fold-changes of all identified proteins in a particular organ. Strong induction of Hp was observed in all tissues accounting for 7% of all proteome changes in the liver, 36% in kidney, 34% in heart, 45% in brain, and 23% in WAT (Fig. 6d). This is clearly shown in the Hp ribbon, which further divides into 5 sub-ribbons that connects back to the protein distribution of each organ. Interestingly, the pattern of the liver was unique because it was largely dominated by liver-specific markers, as opposed to the other organs where the largest fold-changes were instead associated with the shared vascular proteome. Proteoglycan 4 (Prg4), accounted for 34% of all liver vascular surface proteome alterations but remained undetected in the other tissues, except for the heart where it corresponded to 1% of the cardiac proteome changes.

Furthermore, we included a more explanatory legend to Figure 6:

Response: “Circos plot depicting the normalized fold-changes of the top 50 differential proteins across five organs. Each protein value is expressed as a ribbon, the width of which corresponds to the normalized fold-change of that protein as a percentage of the summed fold-changes of all identified proteins in each tissue. Haptoglobin (Hp) is marked with an encircled number 1 to illustrate proteins displaying large induction in all tissues, whereas proteoglycan 4 (Prg4) is marked with an encircled number 2 to highlight proteins displaying very large fold-changes in a tissue-specific fashion.”

Validity: At its current stage the manuscript has several flaws that need to be addressed before publication. This concerns aspects of data reproducibility, inconsistent usage of experimental data sets, incomplete methods and figure legends (all detailed below).

Response: We agree on all points and thank the reviewers for the impartial reviews. We believe all issues raised by the reviewers have been addressed and amended.

Reproducibility: What I miss in this paper are clear statements on the intraassay and interassay variability, an assessment of its quality and discussion of potential impact factors during sample generation and processing. The intraassay variability (animals from the same experimental group) is described in detail (lines 234-250). However, using the term “intraassay variability” will help the reader to understand what was compared here (diff. animals or different experimental groups). Moreover, what I miss here is a statement on the quality of the intraassay variability. Importantly, the interassay variability in terms of protein identifications is briefly mentioned in line 235, but I cannot comprehend how the mentioned 20% reproducible proteins were calculated. This is discussed in detail below.

Response: This major concern raised by the reviewers has been addressed in the following way:

1) We added a fourth experiment to the manuscript and combined all datasets reflecting proteomics analysis of 5 different organs from n=12 infected and n=12 uninfected mice. We used the combined dataset to look at the variability of the assay and demonstrated that with a new Supplemental Figure 4 depicting profile plots for the significant proteins across the organs. As clearly depicted in this picture, there is medium-high intrassay and interassay variability in the method, but proteome changes during sepsis are larger than this experimental error. Plotting some of the top differential proteins indicated that the method can still detect clear-cut differences related to infection.

2) We applied a new cutoff to the results based on the presence of at least 75% of valid values (in other words, no missing values) in at least one of the groups (uninfected+biotin, infected+ biotin, or PBS controls). In this way we focused on the major differences, although potentially low-abundant interesting hits might have been excluded.

3) In addition, it needs to be clarified that there is always a tradeoff between coverage (i.e. how many proteins that can be detected) and amounts (i.e. how many proteins that can be quantified) when working with MS-based proteomics tools. One of the main reasons for that has to do with the stochastic nature of MS analysis in data-dependent acquisition mode (technical issue). Here we opted for a label-free quantification approach somewhere in the middle between pure discovery proteomics approaches (where identification and NOT quantification is the main objective) and targeted approaches (where highly sensitive quantification of ONLY a limited number of proteins is the main objective).

It would also be of interest to see how well the LFQ-normalized intensities (averaged per experimental group) correlate between the different experimental groups.

Response: We have now added a new Supplemental Figure 4 illustrating this aspect of the data.

Can the PCAs for ALL individual samples (9 infected vs 9 controls) still separate infected from non-infected animals? Such information should be added to the manuscript.

Response: Yes, they can. An updated Figure 5 includes now the results for the combined dataset

Finally, I would suggest moving the final figure (Figure 8a) to the start of the MRSA chapter (after line 233). Please show CFU data for all three experiments (ideally in the same graph), if you have them. This will illustrate the intra- and interassay variability of the infection model. Moreover, please provide disease severity scores (if available).

Response: This was a very helpful suggestion. In fact, we decided to make a completely new figure (Figure 4) that includes the information of the former Figure 8a, and discuss the infection model as a separate section.

Inconsistent usage of experimental data sets: A major drawback of the manuscript at its current state is the inconsistent usage of experimental data sets. Throughout the paper the authors seemingly randomly switch between showing data for a “representative data set” with n=3 per group or pooled data for all three data sets with n=9. For instance, after the authors just explained that their experimental groups showed “good correlations” (line 239, Figure S4), I was surprised to see in the key figure of the whole manuscript (figure 4) only volcano plots, PCA etc. for one experimental group (3 infected vs. 3 non-infected animals). In the subsequent figures 5 and 6 it seems like the complete data set was used again. Do the authors have good reasons to show only one experimental data set instead of all data? If not, I strongly suggest to generally show the data for all three experimental groups to increase group size to 9 and hence reduce the β error.

Response: We have now added a fourth experiment to the manuscript and combined all datasets reflecting proteomics analysis of 5 different organs from n=12 infected and n=12 uninfected mice. All files were analyzed as a combined dataset and all figures, tables, table legends, and main text body have been updated with this new data.

Missing information: Several figure legends are incomplete. Information on group sizes and whether the graphs show a representative subset or the full data set are frequently missing in the figure legends (e.g. Figure 2, 3, 4, 7, partially 8).

Response: All figure legends have now been expanded to include the required information

Moreover, the listed methods are incomplete: The following methods are missing: hyaluronic acid determination in plasma (Fig 7g), CFU determination (Figure 8a), Quantification of alanine aminotransferase (ALT) and aspartate aminotransferase (AST) in plasma (Fig 8b), quantitative PCR for PGR4 (Fig 8i).

Response: A description of the missing methods has now been added to the material and method section of the paper

Finally, I suggest to include a more detailed description of the animal model. If the infection per se is highly variable, this will impact on all subsequent analyses. So if available, please provide CFU data for the animals that underwent biotin perfusion. Did all experimental animals survive the septic challenge?

Response: The infection model is now discussed as a separate section in the manuscript. Unfortunately, organ CFU values are not available for the animals that underwent chemical labeling. Instead, an independent cohort of infected animals was analyzed. We have now added more data points to this organ CFU data to better illustrate the variability of our infection model.

Appropriate use of statistics and treatment of uncertainties: As mentioned above, a major drawback of the manuscript at its current state is the inconsistent usage of experimental data sets. When only 1 data set was used, the group size (n=3) was critically low.

Response: We have now added a fourth experiment to the manuscript and combined all datasets reflecting proteomics analysis of 5 different organs from n=12 infected and n=12 uninfected mice. All samples were analyzed as a combined dataset and all figures, figure legends, tables and main text have been updated with this new data.

Suggested improvements: The key knowledge gain from this paper is the core and organ-specific vascular proteomes in healthy and septic mice. Unfortunately, the authors describe the core and organ-specific vascular proteome of healthy mice only for liver and kidneys (figure 3, S2, lines 161ff), even though they have obtained data from noninfected control mice from five organs in the following experiment. It would be a great improvement to the manuscript to use these data to define the core and organ-specific vascular proteome under physiological conditions. In consequence, the current part comparing only liver and kidney would be replaced (subchapter starting at line 161, figure 3, S2).

Response: The major focus of the manuscript is the proteomic alterations taken place at the vascular surfaces during sepsis. The data for the baseline core proteome requested by the reviewer is already included in the manuscript main tables, and is additionally discussed in the context of the deep fractionation (2D-LC-MS/MS) experiments. 2D-LC-MS/MS doubles the number of identifications compared with single runs (LC-MS/MS). However, deep fractionation is cumbersome and cannot be interfaced with the automated sample preparation platform we employed here. For quantification purposes we focused on the septic response using a single step chromatography run, even if this means smaller number of identifications.

References: references are correctly cited. Completeness cannot be judged by me. I suggest to add a reference on the new sepsis definition sepsis-3 (Singer et al, JAMA 2016, 315(8): 801–810) as mentioned below.

Response: Suggested reference has now been added.

Clarity and context: The abstract is clear, but a summary sentence is missing. Introduction and conclusions are appropriate.

Response: A summary sentence for the abstract has now been added:

“Collectively, the data indicates that MRSA-sepsis triggers extensive proteome remodeling of the vascular cell surfaces, in a tissue-specific manner”.

Parts outside the scope of my expertise: I have a basic knowledge of proteomics but cannot judge the suitability of the used LC/MS methods and affiliated data analyses and statistics.

Novelty: The in vivo biotinylation approach per se has been previously described. The novel aspect of this manuscript is to embed this technique into an automated workflow for protein sample preparation and protein analyses. This provides the science community with a valuable and innovative tool to study the

vascular proteome by in-vivo labelling in animal models. The described methods will they be of strong interest to others in the community and the wider field.

Response: We agree and thank the reviewer for his/her positive feedback on the overall goal of this study

Potential future applications: Whether the mouse-derived potential sepsis markers in the vascular proteome could have a predictive value in the clinical setting is unsure, since the vascular proteome is not accessible in patients. Maybe circulating glycocalyx proteins might constitute good markers of general or organ-specific pathologies if they reach sufficient levels in plasma, but this has not been addressed experimentally in this study.

Response: We are already working on addressing this interesting issue

Comments on the main text in chronological order:

Line 83: I suggest referring to the most recent definition of sepsis (so-called Sepsis 3.0) in the article (Singer et al, JAMA 2016, 315(8): 801–810). The key element of the new sepsis definition is multi-organ failure, which is also the focus of this work. Hence, I suggest adding the sepsis 3.0 article to ref 12 and 13.

Response: Reference has now been added.

Line 189-199: This paragraph mentions numerous proteins that were enriched in the vascular proteome of either liver or kidneys. However, no reference to any figures/tables is provided. Moreover, no mathematical/statistical approach for selecting the named proteins was mentioned. How were these candidates selected?

Response: Protein identifications for this experiment are presented in supplemental File 1. A reference to that table has now been added to the text. The bioinformatics treatment of the proteomics data is detailed in the method sections under Mass spectrometry analysis and Statistical analysis sections. However, a clarifying paragraph in the text has now been included :

“Matching control tissue from PBS-perfused animals was run through the same analytical pipeline to account for background signals (i.e. sticky non-biotinylated proteins that unspecifically interact with the streptavidin beads). Only peptides displaying at least a 2-fold enrichment compared to the PBS background of the respective organ were considered for further analysis. To add more stringency, a positive protein identification required a minimum of two unique peptides with two MS/MS scans each. The complete bioinformatics analysis was performed as detailed in the Methods section.”

Afterwards, datasets were analysed “more systematically” according to the authors (protein-protein associations using STRING). In this list, some of the previously mentioned candidates are suddenly missing (e.g. renin-angiotensin system for kidneys, scavenger receptors for liver). Is this due to differences in defining protein clusters or due to technical discrepancies?

Response: The network analysis was done by parsing all protein identifications through the STRING database but keeping only highly confident protein associations (STRING association score > 0.07). Therefore, protein-protein interactions that did not pass this first criterion were not considered for further analysis. Additionally, this network was processed by the Louvain clustering algorithm and only highly modular clusters were finally selected and subjected to functional enrichment analysis. This means that the data was rigorously parsed through very stringent criteria, which resulted in the mapping of roughly, 31% of the kidney and 33% of the liver proteins, which clustered into 5 and 4 distinct Louvain communities. Although not perfect, STRING is one of the most heavily used community tools for biological network analysis and it is manually curated and updated with cross database entries as well as with input from the literature. The main point of this section is to highlight the fact that many of the identified proteins operating at the vascular surfaces are coordinated within large protein-protein

interaction networks. Potentially, this indicates that specific mechanisms of proteome regulation might be in place at the vascular surfaces, the extent of which remains to be elucidated.

Is the upper section (lines 189-199) necessary at all, if you perform the more systematic approach anyway?

Response: The network analysis is not only systematic but also more specific since it only focuses on protein-protein interactions. We changed the following sentence for clarification:

“Finally, a network approach was applied to exclusively focus on potential protein-protein associations (PPA).”

Line 234: Please provide some details here: were the absolute protein numbers similar for infected vs. non-infected mice? How many proteins did you detect in non-biotinylated control animals (i.e. background signal)?

Line 235ff: You mention that “Typically, the proteomics screening identified hundreds of proteins from each individual organ. Roughly ~20% of these protein identifications were robust and were consistently identified across all experiments.” First of all, how did you calculate these 20%? Based on the numbers provided in Figure S3 I calculate that between 7.9 and 15.2% of the proteins identified in either of the 3 experiments were detected in all three experiments. Wouldn't a percentage based on the number of common proteins/total no of proteins from ONE experiment make more sense? Based on this, 30-50% of the proteins detected within one experiment were also found in the other two biological replicates. This rate still seems very low. Please explain what factors might influence reproducibility.

Response: We have now added a fourth experiment to the manuscript and combined all datasets reflecting proteomics analysis of 5 different organs from n=12 infected and n=12 uninfected mice. All samples were analyzed as a combined dataset and all figures and tables have been updated with this new data.

Line 283ff: What you describe here (i.e. the proteome signature shared by all organs) is in other contexts often called “core proteome” or “common proteome” (e.g. in line 292). Using such key words (pan proteome, core proteome, variable/organ-specific proteome) would help the readership to extract the main messages from the results section.

Response: We have tried to be consistent in the revised manuscript and call it the shared proteome

Lines 314ff: This paragraph contains an extensive description of proteins involved in angiogenesis that were upregulated in the liver. However, I miss a reference to original data (Figure, tables, supplement?) Can I find data on the mentioned proteins and their tissue-specific expression somewhere in the supplement? If not, I suggest including such a list.

Response: This issue does not apply anymore since we have now added a fourth experiment to the manuscript and combined all datasets reflecting proteomics analysis of 5 different organs from n=12 infected and n=12 uninfected mice. All samples were analyzed as a combined dataset and all figures and tables have been updated with this new data.

Line 354: Your data do not “verify that MRSA is rapidly cleared from the blood circulation and redirected to the liver” since you did not determine blood counts. Please rephrase.

Response: This sentence has now been deleted

Line 499ff: Please specify the type of cage used to house the mice (IVC, open top,...). Where mice housed under SOPF or SPF conditions? Where were the BL/6 mice obtained from? Please check the S. aureus colonization status of your mice (US vendors provide this information in the health sheet). This is

important, because colonization and hence immunological priming might impact on the disease course. Did you have to submit an application for your animal experiments to local authorities? If so, please provide the application ID.

Response: Missing information has been added to Material and Methods

Line 524: Were the isolated proteins immediately analysed or frozen for later analyses?

Response: Samples were frozen at -80°C until further analysis. This information has now been added to the text.

Comments on figures and figure legends:

Figure 1: What is the abundant pink roundish symbol in the upper panel (blood)? If it is of relevance to the assay, you might want to explain it in the figure legend.

Response: It means abundant plasma proteins. A key has now been added to Fig. 1 to explain the symbols.

Legend: The term “quenching” usually describes “any process which decreases the fluorescence intensity of a given substance”, e.g. by excited state reactions, energy transfer, complex-formation and collisional quenching. (source Wikipedia). To my understanding, what you describe here as quenching is simply a washing step which removes the unbound biotin or Sulfo-NHS-biotin. If this is correct, please change the wording here as well as in M&M (line 516) and within figure 1.

Response: Quenching is also a standard term in Chemical Sciences that means stopping a reaction. In this particular case, the NHS-functionality will react with primary amines such as those present in lysine residues or at the very N-terminus of proteins. Tris-buffers act as a quenching agent since it reacts with NHS.

Figure 4: It should be mentioned in the legend that this figure is based on data from one experimental group (n=3).

Response: This issue does not apply in the revised manuscript because we have now added a fourth experiment to the manuscript and combined all datasets reflecting proteomics analysis of 5 different organs from n=12 infected and n=12 uninfected mice. All samples were analyzed as a combined dataset and all figures and tables have been updated with this new data.

Figure 5a: I never came across Circos plots in this context (proteomics) before and have difficulties grasping their content. Please explain them in more detail in the figure legend. What does the colour code in the outer ring mean (left side of the circus plots)? Give an example on how to read the graph (percentage – ribbon width). Should the label for the percentage values be “fold change of each protein as a percentage of the summed fold changes for all proteins” rather than “fold change”? Simply fold change implies the “common” fold change without normalization and is hence confusing.

Why did you normalize the fold-changes at all and not display the real values? The true fold-changes are nowhere mentioned in the manuscript, but would give the reader a very good estimate on how strong the observed changes are.

Response: Figure 5 has been substituted with a new Figure 6. We integrated all individual circos plots into one single image that summarizes the most significant protein hits and how they distribute across the tissues. Additionally, we added the following explanatory paragraph to the main text body:

“The fold-changes associated with the top 50 differential proteins in each tissue were ranked and visualized as Circos plots (Fig.6d). In this representation, each protein corresponds to one ribbon, and the width of the ribbon indicates the normalized fold-change of an individual protein as a percentage of the summed fold-changes of all identified proteins in a particular organ. As mentioned before, strong induction of Hp was observed in all tissues accounting for 7% of all proteome changes in the liver, 36%

in the kidney, 34% in the heart, 45% in the brain, and 23% in the WAT (Fig.6d). This is clearly appreciated by the Hp ribbon, which further divides into 5 sub-ribbons that connects back to the protein distribution of each organ. Interestingly, the pattern of the liver was unique since it was largely dominated by liver-specific markers, as opposed to the other organs where the largest fold-changes were rather associated with the shared vascular proteome. One of them in particular, Proteoglycan 4 (Prg4), accounted for 34% of all liver proteome alterations but remained undetected in the other tissues, except for the heart where it corresponded to 1% of the cardiac proteome changes.”

Furthermore, we included a more explanatory legend to Figure 6:

“Circos plot depicting the normalized fold-changes of the top 50 differential proteins across five organs. Each protein value is expressed as a ribbon, the width of which corresponds to the normalized fold-change of that protein as a percentage of the summed fold-changes of all identified proteins in each tissue. Haptoglobin (Hp) is marked with an encircled number 1 to illustrate proteins displaying large induction in all tissues, whereas proteoglycan 4 (Prg4) is marked with an encircled number 2 to highlight proteins displaying very large fold-changes in a tissue-specific fashion.”

A maybe antiquated suggestion: Couldn't you display the same information in a table with your top protein candidates and their fold change?

Response: We could but it would take too much space

Please indicate which dataset these circos plots are based on (1 or all 3 experiments).

Response: The circos plot includes the combined dataset, which is also stated in the main text and in the new figure legend

Figure 5b and also line 283f: Please mention which data are included here (“proteins consistently identified in each tissue”). Are these infected mice only, or are the data based on a ratio of infected vs non-infected mice?

Response: This issue does not apply anymore in the revised manuscript because we have now added a fourth experiment to the manuscript and combined all datasets reflecting proteomics analysis of 5 different organs from n=12 infected and n=12 uninfected mice. All samples were analyzed as a combined dataset and all figures and tables have been updated with this new data.

Figure 5c: please explain what is meant by “heat maps showing the frequency for each protein...” (Line 671). Same line: “resulted in a significant change between infected and noninfected mice...”

Response: This issue does not apply anymore in the revised manuscript because we have now added a fourth experiment to the manuscript and combined all datasets reflecting proteomics analysis of 5 different organs from n=12 infected and n=12 uninfected mice. All samples were analyzed as a combined dataset and all figures and tables have been updated with this new data.

Figure 7: Title should be changed to “proteome changes in the hepatic vasculature are associated...”

Response: This issue does not apply anymore in the revised manuscript we have now added a fourth experiment to the manuscript and combined all datasets reflecting proteomics analysis of 5 different organs from n=12 infected and n=12 uninfected mice. All samples were analyzed as a combined dataset and all figures and tables have been updated with this new data.

Are the animals in figure 7 a-f identical with one of the experimental groups from the previous figures? If so, why not show the data for all groups? Please mention the chosen data set in the figure legend. Moreover, why are there unequal numbers of uninfected vs. infected mice (3 vs. 7) in Fig 7g? Again, are these mice from the previous experiments or do they belong to a separate experiment?

Response: The systemic perfusions are generally incompatible with cardiac puncture, so the blood chemistry and HA-data was done on a separate cohort. The same applies to the CFU analysis. This has now been clarified in the legend:

“Figure 7. Proteome alterations in the surface of septic liver vasculatures is associated with changes in hyaluronan and hyaluronan-binding proteins. Relative label-free quantification (LFQ) analysis of proteomic changes in the hepatic vasculature during sepsis reveals differential abundance of multiple targets involved in hyaluronic acid recognition (a). The levels of circulating hyaluronic acid in plasma at 24h post-infection were also significantly increased in a separate cohort of infected animals (n=7) compared with controls (n=3) (b). MRSA infection increases expression and deposition of Prg4 along the central veins and the sinusoidal microvasculature (c-d). Prg4-immunoreactivity was also found at the edges of large necrotic thrombi in association with Ly6G+ neutrophils (e-f). qPCR analysis in a separate cohort of mice demonstrated increased expression of hepatic Prg4-mRNA levels already at 12 hr post-infection (infected livers n=7, uninfected controls n=3) (g).”

Figure 8: I suggest moving Figure 8a to the beginning of the sepsis chapter (see comment above), because these data show the reproducibility and variability of the infection model. Are these the same mice as for the proteomics data or a new lot? Please indicate in the figure legend. If you have more than 1 data set, please show all data! If disease severity scores were documented, they could be included as well.

Response: We thank the reviewer for this helpful suggestion. In fact, we decided to make a completely new figure (Figure.4) that includes the information of the former Figure 8a, in addition to discussing the infection model as a new separate chapter.

Figure 8b: Do you have ALT and AST values for more than three animals? Methods for ALT and AST measurement are missing in M&M section.

Response: We have now added more data point to the ALT/AST graphs and added a description in Material and Methods section

Figure 8e-f: I suggest to separate e&f from g&h. Please add PRG4 as image title to e&f. In addition, could you generate an overlay of g and h?

Response: The figure has been updated

Figure S3: Please explain in the figure legend what exactly is shown here. Are these the identified proteins for the infected animals or for the non-infected controls? How do you calculate the no. of proteins from three animals (within one experimental group)?

Response: This does not apply anymore because that picture was removed

Figure S4: You show correlation data for liver. Do you get similar results for kidneys? Please briefly mention this (without necessarily showing the data) in the results section (line 234ff).

Response: Yes, we have now made a comment in the text:

“Similar results were found in other tissues indicating that the identity of the proteome accessible through this analytical strategy differs between systemically biotinylated and control tissues. More importantly, these findings also indicate that the methodology capture differences specifically associated with infection.”

Figure S5: I suggest to specify your groups infected, noninfected and PBS control as Infected + biotinylation; noninfected + biotinylation, uninfected w/o biotinylation.

Suppl table 2: needs a data dictionary and a format that allows reading the headings (i.e Excel). Values within one column should be displayed with the same decimal place.

Response: All tables are now presented as excel files.

Minor comments:

Line 73: change native immunity to innate immunity.

Changed

Line 76: change disease associated molecular patterns to damage associated molecular patterns.

Changed

Line 97: You might change the sentence to “Proteomic approaches have demonstrated changes in the glycocalyx...” to stress that the cited papers used proteomics.

Changed

Line 113/114: It would make sense to point out that the mentioned new techniques allow for semiquantitative analyses.

Changed to : These instruments facilitate high-resolution measurements of fragment ions, improved proteome coverage, lower false discovery rates, and more robust absolute and semiquantitative proteome analysis

Line 188/189: check formatting

Changed

Line 228/229: The description of the animal groups is confusing. Please add: In addition, one non-infected and one infected mouse were perfused... This will make it much easier to understand.

Changed

Line 269: de-enriched = depleted/reduced?

Changed

Line 269: The pattern obtained

Changed

Line 514, 515, 517: “l” for litre should be “L”

Changed

Line 554: 3 mg or protein?

Changed

Line 614, 623: please provide the software versions

Changed

Figure 2: image headings: Isolectin B4 instead of IsolectinB4?

Changed

Line 654: You might mention here that the hypergeometric enrichment test looks for subcellular location of proteins.

Changed

Line 656: change your last “sentence” into a full sentence.

Changed

Figure 3d-e: inconsistent use of capital letters

Changed

Figure S1 legend: line 4: infrared-dye \diamond infrared dye

Changed

Reviewer #3 (Remarks to the Author):

The manuscript by Toledo et al describes a chemical labelling and quantitative mass spectrometry strategy to monitor the dynamic changes of the vascular cell-surface proteome during sepsis. The results indicate that the analyzed organ vasculatures undergo differential remodeling in sepsis. So far, proteome-wide analysis of changes to vascular surfaces under in vivo like conditions during sepsis has remained elusive, partly due to technical limitations. The work by Toeldo et al provides an elegant solution to this problem by specifically quantifying an enriched fraction of the highly relevant vascular sub-proteome to

monitor organ-specific changes during sepsis and thereby provide insights into the molecular mechanisms of multiple organ failure, a hallmark for sepsis. It can be expected that increased knowledge of the remodeling of the vascular surfaces will contribute to organ-specific changes during sepsis that can in future work facilitate the discovery of markers for the molecular classification of sepsis subtypes. Although, the work presented in this manuscript is impressive, of high quality and of high relevance for the field the following comments should be addressed.

The analysis of the quantitative differences between infected and non-infected animals (Figure 4-6) is at times difficult to follow. It is unclear why only a representative experiment is shown in Figure 4 and why different protein filtration strategies were used for Figure 5 and 6. I suspect that the reason is high variability between the biological experiments? If I understand the text correctly, the authors performed three separate biological experiments with three animals per group and experiment, which would give in total of 9 organs per organ-type. I suggest that the author reanalyze their data by merging the biological replicates into one large data matrix (9 infected/non-infected samples per organ type) and use the combined data set to i) display the number of identified proteins per organ and how frequently a given protein was detected across every organ, ii) use the data from the non-infected animals to objectively classify the proteins into shared and organ-restricted proteins iii) show a heat map of the combined data or part of the data to demonstrate the technical reproducibility between experiments and iv) plot the variance distribution of the identified proteins. One of the strong points of the manuscript is the methodological developments and application of the method to sepsis. However, it is important that readers can judge the performance of the established method and that these results are adequately discussed in the discussion section.

Response: We have now added a fourth experiment to the manuscript and combined all datasets reflecting proteomics analysis of 5 different organs from n=12 infected and n=12 uninfected mice. All samples were analyzed as a combined dataset and all figures, figure legends, tables and main text have been updated with this new data.

Furthermore, it seems more straight forward to use the merged data set with the three biological experiments to calculate statistically differentially changed proteins (p-value corrected for all comparisons) between infected and non-infected animals and display this data in Figure 4 rather than showing the data from a representative experiment. If this is not possible, the authors should at least provide the results from the two other biological experiments in the supplement material in a similar manner as shown in Figure 4. In this context it would be interesting to know which of the significantly induced or repressed proteins are shared between the organs or uniquely confined to one or a few organs. The authors should also show the pca plot and heat maps for the combined data set.

Response: An updated Figure 5 includes now the results for the combined dataset. Also, we include a new supplemental file 3 that contains all shared and organ-specific protein identifications as well as the raw data for their pathway enrichment analysis.

In Figure 3a the authors show TIC graphs of the high-pH peptide fractions. These graphs are not very informative and could be moved to the supplement material. It is unclear why only one replicate was used in this analysis?

Response: The main message from the TIC graphs is that proteins eluting from the high pH fractionation on a C18 column at different acetonitrile concentrations behave chromatographically different when analyzed at low pH, which in turn indicates orthogonality in the separations.

Following text has been added to that section:

“Peptide digests were analyzed through an online 2D-LC-MS/MS workflow at high/low pH, as described in the Methods section, to perform deep fractionation of the samples and to increase proteome coverage”

We have also added some text to the figure legend of this figure to make this point more clear:

“Total ion chromatograms for 5 consecutive fractions from liver (a) and kidney (b) peptide digests are shown, indicating that proteins eluting at high pH and at increasing acetonitrile concentrations from the C18 column display different chromatographic behaviors when separated at low pH, which is consistent with an orthogonal fractionation strategy and deeper coverage in terms of protein identifications.”

This experiment is not exactly quantitative, but the objective was to perform a deep fractionation of a few tissues to have a sense of the scope of the vascular proteome. There is always a tradeoff between proteome coverage (how many proteins can be identified) and quantities (how many proteins can be quantified) using label-free quantification strategies. In principle, deep fractionation is informative but time consuming and that’s why we moved onto a single- separation strategy in the following sepsis study, where quantitative profiling of multiple animals and multiple organs was now the main objective.

In Figure 5 the authors show Circos plots which are a bit difficult to interpret. I suggest that these plots are made larger.

Response: Figure 5 has been substituted with a new Figure 6. We integrated all individual circos plots into one single plot that summarizes the most significant protein hits and how they distribute across the tissues. Additionally, we added the following explanatory paragraph to the main text body:

“The fold-changes associated with the top 50 differential proteins in each tissue were ranked and visualized as Circos plots (Fig.6d). In this representation, each protein corresponds to one ribbon, and the width of the ribbon indicates the normalized fold-change of an individual protein as a percentage of the summed fold-changes of all identified proteins in a particular organ. As mentioned before, strong induction of Hp was observed in all tissues accounting for 7% of all proteome changes in the liver, 36% in the kidney, 34% in the heart, 45% in the brain, and 23% in the WAT (Fig.6d). This is clearly appreciated by the Hp ribbon, which further divides into 5 sub-ribbons that connects back to the protein distribution of each organ. Interestingly, the pattern of the liver was unique since it was largely dominated by liver-specific markers, as opposed to the other organs where the largest fold-changes were rather associated with the shared vascular proteome. One of them in particular, Proteoglycan 4 (Prg4), accounted for 34% of all liver proteome alterations but remained undetected in the other tissues, except for the heart where it corresponded to 1% of the cardiac proteome changes.”

Furthermore, we included a more explanatory legend to Figure 6:

“Circos plot depicting the normalized fold-changes of the top 50 differential proteins across five organs. Each protein value is expressed as a ribbon, the width of which corresponds to the normalized fold-change of that protein as a percentage of the summed fold-changes of all identified proteins in each tissue. Haptoglobin (Hp) is marked with an encircled number 1 to illustrate proteins displaying large induction in all tissues, whereas proteoglycan 4 (Prg4) is marked with an encircled number 2 to highlight proteins displaying very large fold-changes in a tissue-specific fashion.”

In figure 5C the authors demonstrate significantly differences for 32 target proteins that were shared among all organs. I suggest that the authors provide additional fold-change plots or a heat maps with log2 protein intensities or z-scored protein intensities for these proteins across all the analyzed organs.

Response: Figure 5 has been replaced with a new Figure 6. A heat map of the average fold-change of the shared proteins across all tissues has been added (6c) according to the helpful suggestion of the reviewer.

Figure 7: are these samples from a representative experiment? If so why are the number of replicates

different for the plasma HA analysis (Fig 7g)? Please state in the Figure legend if the p-values were corrected for multiple hypothesis testing.

Response: The systemic perfusions are generally incompatible with cardiac puncture, so the blood chemistry and HA-data were done on a separate cohort. The same applies to the CFU analysis. This has now been clarified in the figure legends

The authors state on line 336 that the analytical constraints were relaxed. The authors should clarify what this exactly means.

Response: This does not apply anymore because we have now added a fourth experiment to the manuscript and combined all datasets reflecting proteomics analysis of 5 different organs from n=12 infected and n=12 uninfected mice. All samples were analyzed as a combined dataset and all figures and tables have been updated with this new data.

The results from the representative experiment in Figure 4 are convincing. It is however surprising that the authors do not detect increased degradation and shedding of the glycocalyx. In contrast, many proteins seem to be more abundant in the infected animals. Some of the increased proteins are abundant plasma proteins that potentially stick to the vascular cell-surfaces, but for other proteins the reason for the increased abundance is less clear. The manuscript would benefit for an extended discussion where reasons behind these results are discussed in more detail. In particular the absence of visible signs of increased degradation and shedding.

Response: There are several explanations for this apparent discrepancy. 1) most shedding has been documented in models of multibacterial sepsis (CLP) or endotoxic shock (LPS). The amount of shedding taking place during monobacterial sepsis (and MRSA sepsis in particular) is on the other hand less clear. 2) We only looked at a single time point (24 hr), which may not reflect the whole picture since both sepsis and vascular remodeling are dynamic processes. We also cited several relevant papers in the text

Minor comments

- Several of the figure legends are too brief. In Figure legend 4 the authors should add information stating which part of the data that was included in the analysis and provide information of the p-value correction test. In Figure legend five the authors should add text how the proteins were selected for the Circos plots.

Response: All figure legends have now been expanded to include the required information

- Supplementary figure 6d seems have different magnifications between infected and non-infected animals.

Response: All figures include a magnification bar

- Figure 8a, for the sake of consistency, Fat in the y-axis should be changed to WAT.

Response: This figure (now Fig. 4) has been corrected.

- Line 279 – SSA 2 should be changed to SAA2 to refer to serum amyloid A protein 2.

Changed

- Inter-alpha trypsin inhibitor heavy chain 3 and 4 should be designated as Itih3 and Itih4 in mice.

Changed

- Prg4 or PRG4 should be used consistently to define protein or gene names in the text.

Changed

- Add method section for qPCR

PCR has been added to the Methods section

- Figure 8c and d: Add a representative image from healthy tissue as comparison

Response: Streptavidin staining of a healthy liver is shown in Fig.2. Hematoxylin and Eosin stain of healthy livers is shown in supplemental Fig.3a

Reviewer #4 (Remarks to the Author):

The authors present a surface proteomic atlas of mouse vasculature and how this changes after infection in the context of sepsis. The authors examined several organs and show remarkable depth in their sensitivity of protein detection using the latest spectrophotometry methods. This is a method that may have applicability to a wide audience. Two fundamental elements diminish enthusiasm for the study. The main concern is the reproducibility of the technique. The proteomics only saw 20% of the protein identifications as being robust and consistent across all experiments, as stated in the text. As an example the venn diagram of liver shows that while there were 1,472 proteins identified only 215 (15%) were consistent across the 3 mice, which show a pretty high lack of reproducibility. In practical terms, this would require a large number of mice and resources, in the way of proteomics, to gain a firm idea on what changes are likely to be real.

Response: This major concern raised by the reviewers has been addressed in the following way:

1) We added a fourth experiment to the manuscript and combined all datasets reflecting proteomics analysis of 5 different organs from n=12 infected and n=12 uninfected mice. We used the combined dataset to look at the variability of the assay and exemplified that with a new Supplemental Figure 4 depicting profile plots for the significant proteins across the organs. As clearly depicted in this picture, there is medium-high intrassay and interassay variability in the method, but proteome changes during sepsis are larger than this experimental error. Plotting some of the top differential proteins indicated that the method can still detect clear-cut differences related to infection.

2) We applied a new cutoff to the results based on the presence of at least 75% of valid values (in other words, no missing values) in at least one of the groups (uninfected+biotin, infected+ biotin, or PBS controls). In such a way we focused on the major differences although potentially low-abundant interesting hits might have been excluded.

3) In addition, it needs to be clarified that there is always a tradeoff between coverage (i.e. how many proteins that can be detected) and amounts (i.e. how many proteins that can be quantified) when working with MS-based proteomics tools. One of the main reasons for that has to do with the stochastic nature of MS analysis in data-dependent acquisition mode (technical issue). Here we opted for a label-free quantification approach somewhere in the middle between pure discovery proteomics approaches (where identification and NOT quantification is the main objective) and targeted approaches (where highly sensitive quantification of ONLY a limited number of proteins is the main objective).

The second major point is the sepsis information generate is largely descriptive, hence the paper could be constructed as more of a resource style article.

Response: We agree on the nature of this study being largely descriptive. However, we argue that the information summarized in the manuscript will still be of interest to a broad sector of the scientific community working in the fields of infectious disease, microbiology, vascular biology, proteomics and related areas. The transdisciplinary character of this work is what it makes it suitable for Nature Communication since this journal favored publications at the intersection of multiple fields.

Minor points

The data in figure 5a, might be visualized differently. The different magnitudes of the proteins are not clearly evident in this type of plot.

Response: Figure 5 has been substituted with a new Figure 6. We integrated all individual circus plots into one single plot that summarizes the most significant protein hits and how they distribute across the tissues. Additionally, we added the following explanatory paragraph to the main text body:

“The fold-changes associated with the top 50 differential proteins in each tissue were ranked and visualized as Circos plots (Fig.6d). In this representation, each protein corresponds to one ribbon, and the width of the ribbon indicates the normalized fold-change of an individual protein as a percentage of the summed fold-changes of all identified proteins in a particular organ. As mentioned before, strong induction of Hp was observed in all tissues accounting for 7% of all proteome changes in the liver, 36% in the kidney, 34% in the heart, 45% in the brain, and 23% in the WAT (Fig.6d). This is clearly appreciated by the Hp ribbon, which further divides into 5 sub-ribbons that connects back to the protein distribution of each organ. Interestingly, the pattern of the liver was unique since it was largely dominated by liver-specific markers, as opposed to the other organs where the largest fold-changes were rather associated with the shared vascular proteome. One of them in particular, Proteoglycan 4 (Prg4), accounted for 34% of all liver proteome alterations but remained undetected in the other tissues, except for the heart where it corresponded to 1% of the cardiac proteome changes.”

Furthermore, we included a more explanatory legend to Figure 6:

“Circos plot depicting the normalized fold-changes of the top 50 differential proteins across five organs. Each protein value is expressed as a ribbon, the width of which corresponds to the normalized fold-change of that protein as a percentage of the summed fold-changes of all identified proteins in each tissue. Haptoglobin (Hp) is marked with an encircled number 1 to illustrate proteins displaying large induction in all tissues, whereas proteoglycan 4 (Prg4) is marked with an encircled number 2 to highlight proteins displaying very large fold-changes in a tissue-specific fashion.”

Figure 8 could include a more zoomed out image to show the pathology overall.

Response: We believe the images are adequate to depict the pathology.

REVIEWERS' COMMENTS:

Reviewer #2 (Remarks to the Author):

My comments have all been addressed to my satisfaction

Reviewer #3 (Remarks to the Author):

The manuscript by Toledo et al describes a chemical labelling and quantitative mass spectrometry strategy to monitor the dynamic changes of the vascular cell-surface proteome during sepsis. The major comments in the last round of revisions was related to the use of representative experiments rather than combined analysis of all independent experiments. These comments have now been fully resolved by the authors. Overall, the paper is novel and will be of interest to others in the community and the wider field.

Minor comments

-The authors state on line 276 that they use "two-way analysis of variance (ANOVA) with a permutation-based false discovery rate correction for multiple test comparisons". However, in the method section on line 649 the authors state "Statistically significant changes between groups were assessed by a two-tailed student's t-test using a permutation-based FDR for multiple test correction and truncation after 250 randomizations". The authors needs to clarify why two different tests were used.

Reviewer #4 (Remarks to the Author):

The authors have made substantial efforts to address the previous concerns and now present a much improved manuscript.

REVIEWERS' COMMENTS:

Reviewer #2 (Remarks to the Author):

My comments have all been addressed to my satisfaction

Reviewer #3 (Remarks to the Author):

The manuscript by Toledo et al describes a chemical labelling and quantitative mass spectrometry strategy to monitor the dynamic changes of the vascular cell-surface proteome during sepsis. The major comments in the last round of revisions was related to the use of representative experiments rather than combined analysis of all independent experiments. These comments have now been fully resolved by the authors. Overall, the paper is novel and will be of interest to others in the community and the wider field.

Minor comments

-The authors state on line 276 that they use "two-way analysis of variance (ANOVA) with a permutation-based false discovery rate correction for multiple test comparisons". However, in the method section on line 649 the authors state "Statistically significant changes between groups were assessed by a two-tailed student's t-test using a permutation-based FDR for multiple test correction and truncation after 250 randomizations". The authors needs to clarify why two different tests were used.

-We thank the reviewer for the comment. This is in fact a typo from previous versions and has been updated to:

"Statistically significant changes between groups were assessed by two-way analysis of variance (ANOVA) with a permutation-based false discovery rate (FDR) for multiple test correction and truncation after 250 randomizations".

All ANOVA p- and q-values are in fact stated in Supplemental Data 2

Reviewer #4 (Remarks to the Author):

The authors have made substantial efforts to address the previous concerns and now present a much improved manuscript.